# HAMSTER: Hyperspectral Albedo Maps dataset with high Spatial and TEmporal Resolution

Giulia Roccetti[1,2], Luca Bugliaro[3], Felix Gödde[1], Claudia Emde[3,1], Ulrich Hamann[4], Mihail Manev[1], Michael Fritz Sterzik[2], and Cedric Wehrum[1]

[1]Meteorologisches Institut, Ludwig-Maximilians-Universität München, Munich, Germany
[2]European Southern Observatory, Karl-Schwarzschild-Straße 2, 85748, Garching bei München, Germany
[3]Deutsches Zentrum für Luft- und Raumfahrt, Institut für Physik der Atmosphäre, Oberpfaffenhofen, Germany
[4]Federal Office of Meteorology and Climatology MeteoSwiss, Locarno-Monti, Switzerland

**Correspondence:** Giulia Roccetti (giulia.roccetti@eso.org)

**Abstract.** Surface albedo is an important parameter in radiative transfer simulations of the Earth's system, as it is fundamental to correctly calculate the energy budget of the planet. The Moderate Resolution Imaging Spectroradiometer (MODIS) instruments on NASA's Terra and Aqua satellites continuously monitor daily and yearly changes in reflection at the planetary surface. The MODIS Surface Reflectance black-sky albedo dataset (MCD43D, version 6.1) gives detailed albedo maps in seven spectral bands in the visible and near-infrared range. These albedo maps allow us to classify different Lambertian surface types and their seasonal and yearly variability and change, albeit only in seven spectral bands. However, a complete set of albedo maps covering the entire wavelength range is required to simulate radiance spectra, and to correctly retrieve atmospheric and cloud properties from Earth's remote sensing. We use a Principal Component Analysis (PCA) regression algorithm to generate hyperspectral albedo maps of Earth. Combining different datasets of hyperspectral reflectance laboratory measurements for various dry soils, vegetation surfaces, and mixtures of both, we reconstruct the albedo maps in the entire wavelength range from 400 to 2500 nm. The PCA method is trained with a 10-years average of MODIS data for each day of the year. We obtain hyperspectral albedo maps with a spatial resolution of 0.05° in latitude and longitude, a spectral resolution of 10 nm, and a temporal resolution of 1 day. Using the hyperspectral albedo maps, we estimate the spectral profiles of different land surfaces, such as forests, deserts, cities and icy surfaces, and study their seasonal variability. These albedo maps shall enable to refine calculations of Earth's energy budget, its seasonal variability, and improve climate simulations.

## 1 Introduction

The surface albedo of the planet plays a crucial role within the climate system, governing the proportion of reflected solar light over the incoming solar radiation at the surface. This holds significant importance as it effectively regulates Earth's surface energy budget (Liang et al., 2010; He et al., 2014). The role of albedo extends to climate regulation, with snow and ice albedo feedback exerting a significant influence on climate change dynamics. Snow and ice possess much higher reflectivity compared to the surfaces they overlay. As temperatures rise, the diminishing extent of snow and ice cover leads to a decline in the planet's albedo. Consequently, this intensifies surface warming through a positive feedback mechanism.

Land surface albedo displays remarkable variability, both spatially and temporally. Notable fluctuations in surface albedo coincide with changes in land cover and surface conditions, including factors like vegetation (Loarie et al., 2011; Lyons et al., 2008), snow (He et al., 2013), soil moisture (Govaerts and Lattanzio, 2008; Zhu et al., 2011), and urban development (Offerle et al., 2005). In addition, soil and vegetation surfaces show different reflectance behaviours as a function of wavelength, which are usually not incorporated in Earth system models (ESMs).

In the last decades, the advancement of satellite remote sensing techniques has enabled more accurate monitoring of Earth's surface, enhancing radiative transfer and climate models. This progress allows for continuous acquisition of extensive land surface observation data. However, climate models still struggle to capture albedo's temporal and spatial variations. In particular, global and regional climate models often necessitate albedo products with absolute accuracy of 0.02–0.03 (Sellers et al., 1995; He et al., 2014). Zhang et al. (2010) compared the Moderate Resolution Imaging Spectroradiometer (MODIS) albedo products with the Coupled Model Intercomparison Project Phase 3 (CMIP3) model results spanning 2000 to 2008, revealing discrepancies in globally averaged albedo of up to 0.06. In addition, validation of different satellites land surface products, such as MODIS (Schaaf et al., 2002), GLASS (Global LAnd Surface Satellite, Liu et al. (2013); Qu et al. (2014)), and CGLS (Copernicus Global Land Service, Buchhorn et al. (2020)), shows global absolute differences up to 0.02–0.06, with the largest variation occasionally exceeding 0.1 (Shao et al., 2021).

The divergence of different albedo products is not the only source of uncertainty in ESMs radiative transfer calculations. Most ESMs use a two-stream approach for the land component, where soil albedo has fixed values in two spectral broadband regions: the photosynthetically active radiation band (PAR, 400–700 nm) and the near-infrared band (NIR, 700–2500 nm). However, broadband radiative transfer schemes show a strong spectral discontinuity at 700 nm (Braghiere et al., 2023). This divergence in surface reflectance propagates into other radiative partitioning terms, such as absorptance and transmittance at the top of atmosphere (TOA).

More in general, in cloud-free simulations over land, the dominant factor impacting the TOA visible (VIS) and near-infrared radiance is surface reflection (Vidot and Borbás, 2014). Varied surface optical properties exhibit distinct spectral signatures contingent on the type of surface. Furthermore, within the VIS/NIR range, surface optical properties showcase a robust geometrical reliance that changes according to solar and satellite directions. To elucidate the spectral reliance of the surface, an assumption of Lambertian behaviour can be made, implying isotropic luminance regardless of the viewer's angle. The albedo quantifies the proportion of reflected light under the assumption of isotropic radiation reflection.

Polar-orbiting satellites, like NASA's Terra and Aqua, provide global albedo maps, which are vital for spectral, temporal, and spatial global albedo assessment. The MODIS instrument, on NASA's Terra and Aqua satellites, offers coverage of Earth's surface every 1 to 2 days, enhancing our understanding of terrestrial, oceanic, and atmospheric processes. In the VIS/NIR range, MODIS features seven spectral bands delivering data on land surface characteristics. However, radiative transfer simulations demand precise radiance calculations across all wavelengths, which necessitates hyperspectral albedo maps. For example, retrievals of cloud pressure thickness using the O2A band (760-770 nm), requires precise albedo estimates in this spectral region (Li and Yang, 2024). Such comprehensive data are lacking due to the impracticality of obtaining albedo maps from satellites for every wavelength. As a result, various assumptions are incorporated into radiative transfer codes to overcome this lack of

information.

The MODIS albedo measurements are derived simultaneously from the Bidirectional Reflectance Distribution Function (BRDF), depicting radiation discrepancies resulting from the scattering (anisotropy) of individual pixels. This methodology relies on multi-date, atmospherically corrected, and cloud-cleared input data obtained over 16 day intervals. The spatial resolution is set at 30 arc seconds in latitude and longitude (equivalent to 1 km at the equator) using the Climate Modeling Grid (CMG). To derive climatological averages, the MODIS MCD43D42-48 albedo datasets are averaged over a 10-year span in steps of 1 day, and albedo maps are built for each day.

In this work, we introduce a novel methodology for creating hyperspectral albedo maps based on the seven representative bands of the MODIS instrument. Using a Principal Component Analysis (PCA) regression approach, we combine different soils, rocks and vegetation datasets representative of different parts of the world, along with the Lambertian surface albedo maps from the MCD43D (version 6.1) product (Schaaf and Wang, 2021) derived from the Terra and Aqua satellites. These maps cover the seven bandpasses relevant for land surface albedos. Employing a PCA algorithm, as previously done by Vidot and Borbás (2014) and Jiang and Fang (2019), enables us to reduce the problem's high dimensionality and to generate new albedo maps by interpolating between the measured bandpasses.

These hyperspectral albedo maps of Lambertian surfaces hold significance in various climate and radiative transfer models for Earth's system. Using an ESM with coupled atmosphere-land simulations, Braghiere et al. (2023) demonstrated the impact of making simplistic assumptions on albedo maps, using only two broadband values, compared to hyperspectral albedo maps. They combined the Community Land Model version 5 (CLM5) (Lawrence et al., 2019) soil color scheme with the eigenvectors calculated from the General Spectral Vector (GSV) decomposition algorithm (Jiang and Fang, 2019) to build hyperspectral soil reflectance maps to assess their impact on ESMs. Differently from our dataset of hyperspectral albedo maps, their approach is not based on satellite measurments, so it is less accurate and misses the seasonal and temporal variability of surface reflectance. However, it holds significance in assessing the impact of the hyperspectral treatment of Lambertian albedo in ESMs. Braghiere et al. (2023) estimated a divergence of the radiative forcing of 3.55 W m$^{-2}$, which subsequently impacts the net solar flux at TOA (> 3.3 W m$^{-2}$), cloudiness, rainfall, surface temperature and latent heat fluxes. Braghiere et al. (2023) also highlights the impact of implementing hyperspectral albedo maps in regional models, where differences in latent heat can be higher than 5 W m$^{-2}$, showing the implications for regional climate variability and prediction of extreme events.

In the near future, the launch of new satellite missions like NASA's Earth Surface Mineral Dust Source Investigation (EMIT), will allow obtaining hyperspectral soils and vegetations data and to benchmark the accuracy of the model generated hyperspectral maps.

## 2  Data and Methods

### 2.1  MODIS surface albedo climatology

NASA's MODIS instruments (Salomonson et al., 1989) aboard the Terra and Aqua satellites, launched in 1999 and 2002, observe the Earth in 36 spectral bands. Two channels, centred at 645 and 858 nm (see Tab. 1), have a spatial resolution of

**Table 1.** Spectral bands of MODIS in the VIS and NIR providing information about the land surface. For each band, we specify the central wavelength and the bandwidth.

| band | central $\lambda$ [nm] | bandwidth [nm] |
|------|------------------------|----------------|
| 1 | 645 | 620–670 |
| 2 | 858 | 841–876 |
| 3 | 469 | 459–479 |
| 4 | 555 | 545–565 |
| 5 | 1240 | 1230–1250 |
| 6 | 1640 | 1628–1652 |
| 7 | 2130 | 2105–2155 |

250 m, and five channels (centred at 469, 555, 1240, 1640, 2130 nm), including three in the shortwave infrared, have a spatial resolution of 500 m. All other channels have a resolution of 1 km.

The science dataset MCD43D (version 6.1), (Schaaf and Wang, 2021), is the combined Aqua+Terra L3 MODIS Surface Reflectance product and provides daily global estimates of directional hemispherical surface reflectance (black-sky albedo) and bihemispherical surface reflectance (white-sky albedo) for the seven (2+5) MODIS bands mentioned above, and for three broadband spectral intervals (visible 300–700 nm, near-infrared 700-5000 nm and shortwave 300–5000 nm) with a spatial resolution of 30 arc seconds in latitude and longitude (corresponding to roughly 1000 m at the equator). MODIS cloud-free observations are collected over 16 days and corrected for atmospheric gases and aerosols to derive surface albedo for land pixels (water bodies are not considered). Data are temporally weighted to the ninth day of the retrieval period, and this day appears in the file name. Each surface reflectance pixel contains the best possible measurement of the period, selected on the basis of high observation coverage, low view angle, the absence of clouds or cloud shadow, and aerosol loading. Usually, due to the sun-synchronous orbit of the Terra and Aqua satellites (equatorial crossing times at 10:30 AM and 1:30 PM respectively), only pixels with local solar noon zenith angle up to approximately 80° are provided with an albedo value.

The MODIS land surface products have been validated against in situ measurements and other satellite-based land surface albedo. Globally, the MODIS product is less accurate for high solar zenith angles (Sánchez-Zapero et al., 2023). We compile a black-sky albedo climatology for the seven MODIS spectral bands starting from the MCD43D42-48 products. We average the daily available MODIS product over a 10 years span, from 2013 to 2022, in steps of 1 day, starting on January 1st, i.e., from day of the year (DOY) 1 to DOY 365. This results in 365 climatologically averaged albedo maps for spectral band with a spatial resolution of 30 arc seconds in latitude and longitude. The aim is a complete surface albedo climatology map for all grid boxes that are illuminated by the Sun, i.e., up to a local solar noon zenith angle of 90°. Pixels that are in the dark (i.e., the

Sun is always below the horizon) during the entire DOY are left unfilled. For the computation of the climatology, we proceed in the following way:

1. First, we select the MCD43D42-48 albedo retrievals with albedo quality between 0 and 3 (see Tab. 2) and compute the mean value of the surface albedo for every grid box over the 10 years for a given DOY. After this averaging procedure, some pixels remain unfilled due, e.g., to cloudiness and because of the local solar noon zenith angle constraints mentioned above.

2. Thus, for every DOY we fill the missing values with the mean of the albedo at DOY-$n$ and DOY+$n$ (temporal averages obtained in 1), with $n \in [1, 40]$. The mean value with the smallest $n$ is the one that is used, i.e., the value that is closest in time.

3. For some DOYs, close to solstices and for local solar noon zenith angle between 80° and 90°, a range of 40 days is not enough to have a filled value both in the future and in the past. It might be, for example, that a value is available close in the future, but to have a corresponding value in the past we should look further than 40 days. The reason why we require, in the previous step, to have values both in the past and in the future is to balance out seasonal changes and avoid sharp transitions near the solstices. In this cases, we first search for the closest filled values both in the past and in the future, even if the two intervals are different or if one of them is larger than 40 days. Then, we average the values of albedo in a 10-days interval around the selected future and past available days. Instead of simply assigning to the actual DOY the mean of these averages, we perform a linear interpolation, to weight more the values closer in time to the actual DOY.

4. In a forth step, remaining missing values for a given DOY are replaced with the spatial average for the same DOY over an area of $m \times m$ grid boxes around each missing value, with $m \in [3, 5, 7, 9]$. The mean value with the smallest $m$ is the one that is used, i.e., the value corresponding to the smallest surrounding area.

5. Further remaining missing values are replaced with the mean over longitude of surface albedo in 2° latitude bands for the same DOY.

6. If missing values still exist at this stage for given grid boxes and a given DOY, the mean value over all DOYs during the 10 years under consideration is used to replace them.

7. Finally, since the MCD43D product only retrieves land properties, we compute an albedo value for the ocean pixels in each of the seven MODIS bands using the "deep ocean" spectrum from the old ECOSTRESS library of the US Geological Survey (USGS) database (Baldridge et al., 2009; Meerdink et al., 2019). To this end, incoming solar spectral irradiance (Kurucz, 1992) is first convolved with the spectral response function of the given MODIS channels. Then, under the assumption of no atmosphere, reflected spectral irradiance at the surface is computed upon multiplication with the spectral ocean albedo and integrated over wavelength. This value is finally divided by the integral of the incoming spectral irradiance computed above to obtain the band albedo values for the ocean. These values are used everywhere

**Table 2.** Meaning of the MCD43D albedo quality flag.

| Flag value | Description |
| --- | --- |
| 0 | Best quality, full BRDF inversions |
| 1 | Good quality, full BRDF inversions |
| 2 | Magnitude inversion (number of observations $\geq 7$) |
| 3 | Magnitude inversion (number of observations $\geq 2$ and $< 7$) |
| 255 | Fill value |

for the global water bodies and at every time. Of course, we are aware that water surfaces are better characterised using a BRDF in order to take care of specular reflection (Cox and Munk, 1954a, b; Nakajima, 1983).

MODIS provides data also over coastal regions covering some ocean pixels. These pixels were filled in the climatology as in
steps 1-6, and not replaced as ocean pixels by step 7. Some of these coastal pixels also exhibit sea ice, which remains included also in the climatology.

The percentage of missing land pixels filled after each step of the climatology is shown in Fig. 1. The percentage is calculated

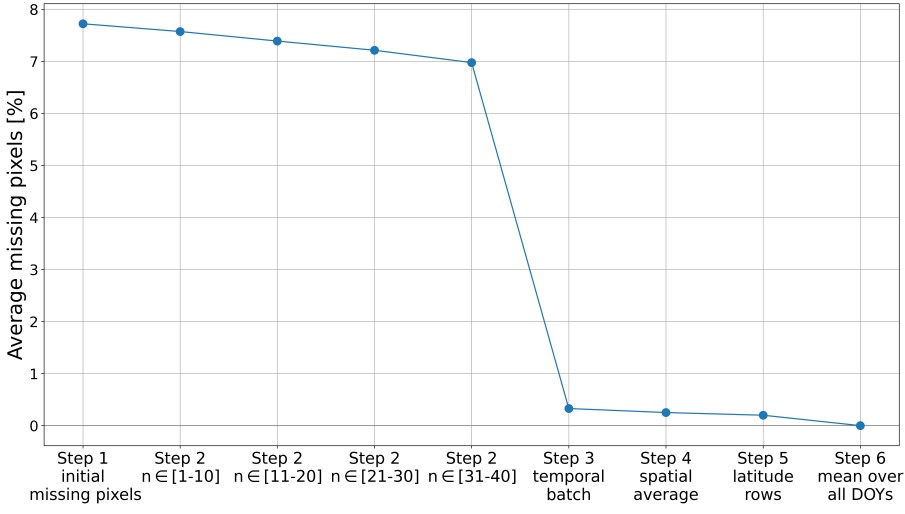

**Figure 1.** Percentage of land missing pixels as the average over all DOYs. We specify which is the remaining percentage of missing values after each step of the climatology.

as the average over all DOYs. Step 3 fills most of the remaining missing pixels, but only between 80° and 90° of local solar noon zenith angle. These pixels receive only almost parallel incoming solar radiation, and thus their impact on radiative transfer
calculations is limited. On the other hand, our methodology allows to give an estimate to these high local solar noon zenith angle pixels, which are usually also highly reflective in the visible wavelengths.

This climatology is the starting point to build the hyperspectral albedo maps, where the average ice and snow cover is automatically included. Our MODIS black-sky albedo climatology from years 2013 to 2022 is available at https://doi.org/10.57970/pt52a-nhm92. For every pixel, we provide a flag indicating at which step the albedo value was filled. The spatial resolution is the same as the MCD43D product (30 arc seconds).

## 2.2 Soils and Vegetations spectra

To create the hyperspectral albedo maps for each DOY, we use laboratory and in-situ hyperspectral measurements of different soils, rocks and vegetation surfaces. Jiang and Fang (2019) developed hyperspectral soil reflectance eigenvectors to improve canopy radiative transfer. Studying the impact of different regional datasets, they found an increase of accuracy and robustness with a global sample coverage of different soils and vegetations spectra, compared to the performances of regional datasets. Following this prescription, we select three dry soils and vegetations datasets, covering different countries and different surface materials:

1. the ECOSTRESS library (Baldridge et al., 2009; Meerdink et al., 2019) includes 1023 surface spectra from the United States, among which 487 are vegetations spectra, 62 non-photosynthetic vegetations, 381 rocks, 40 soils, 45 man-made materials, and 8 are water ice and snow spectra;

2. the ICRAF–ISRIC dataset (ICRAF-ISRIC, 2021), which is a global dataset with 4440 spectra of different soils from 58 different countries (spanning Africa, Asia, Europe, North America, and South America);

3. the LUCAS dataset (Orgiazzi et al., 2018), which contains 21782 different soils spectra from 28 European Union countries (including the United Kingdom), where we selected the 30° viewing angle. As shown by Shepherd et al. (2003), LUCAS spectra are problematic between 400 and 500 nm, where they reach negative values. Following Jiang and Fang (2019), we use the multiple linear regression algorithm from scikit learn (`sklearn.linear_model.LinearRegression`) (Pedregosa et al., 2011), trained on the ISAC–ISRIC dataset to reconstruct the LUCAS spectra in the 400 to 500 nm spectral range.

All the datasets cover the 400–2500 nm spectral range, with different spectral resolutions. LUCAS dataset has a spectral resolution of 0.5 nm, while ICRAF–ISRIC and ECOSTRESS of 10 nm. We interpolated the least resolved datasets to obtain all spectra with a resolution of 1 nm. Among the water bodies in ECOSTRESS, there are three different snow spectra: coarse granular snow, medium granular snow and fine snow. In addition, there are also frost and ice spectra, sea foam, sea water and tap water. All together, they form the 8 water and snow spectra mentioned for the ECOSTRESS library.

In total, we use 26635 dry soils, vegetations, snow and ice spectra from 82 different countries as input to extract the principal components. In Fig. 2, we show some representative soil and vegetations spectra from the ECOSTRESS library. One limation of our approach relies on the fact that vegetation spectra are only present in the ECOSTRESS library, which is a local dataset from the United States. However, to our knowledge, this is the only available datset showing trees, shurbs and grass spectra which are fundamental for our purpose. Jiang and Fang (2019) also study the influence of humid soils on the PCA regression

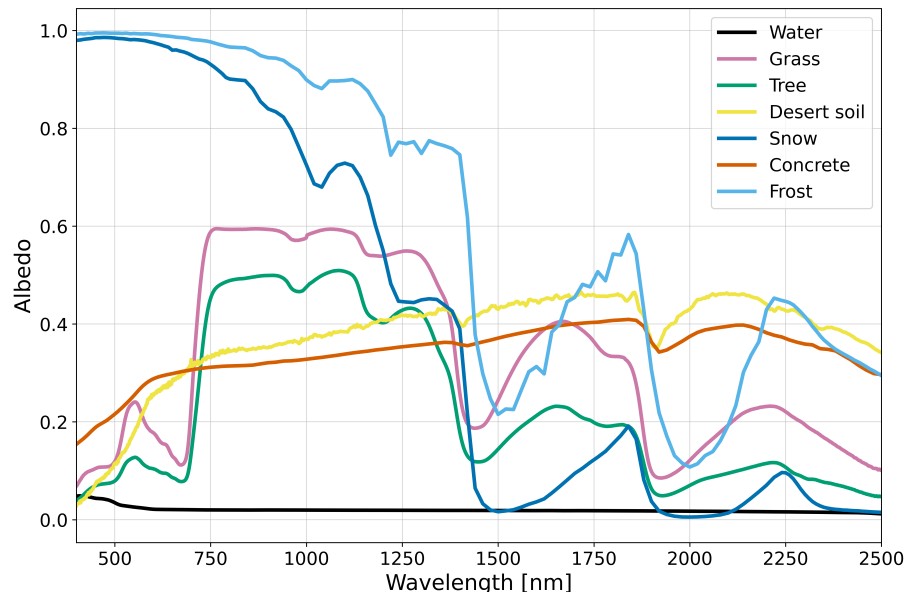

**Figure 2.** Albedo spectral signatures of some typical soils, vegetation and water bodies from the ECOSTRESS library.

algorithm. They find that the effect of soil moisture is non-linear, causing a general reduction of reflectance due to a total internal reflection effect of the water surface. This effect is more prominent in the near infrared (1100–2500 nm). They conclude that to treat the dry and humid soils separately leads to a more applicable soil reflectance model. A comprehensive and global database of humid soils is currently not available in the literature, and the inclusion of humid soils is outside the scope of our work.

### 2.3 Principal Component Analysis

The vector of the MODIS albedo data (Sect. 2.1) at the seven wavelengths ($\boldsymbol{R}$) can, in general, be decomposed as

$$\boldsymbol{R} = \boldsymbol{c}\mathbf{U}, \tag{1}$$

where $\boldsymbol{R} = (r_1, ..., r_n)$ is the albedo vector, with $n$ the number of wavelengths, $\boldsymbol{c} = (c_1, ..., c_m)$ is the coefficient vector, with $m$ representing the number of surface spectra and $\mathbf{U}$ is an $m \times n$ matrix with the laboratory spectra of different soils and vegetation types. In order to calculate the hyperspectral albedo maps, we first need to compute the coefficient vector $\boldsymbol{c}$ at every pixel, by inverting Eq. 1. Since the $\mathbf{U}$ matrix is not square, the correct inverse equation is:

$$\boldsymbol{c} = \boldsymbol{R}\mathbf{U}^T(\mathbf{U}\mathbf{U}^T)^{-1}. \tag{2}$$

From the MODIS dataset, $\boldsymbol{R}$ is available only for seven spectral bands (see Tab. 1), while the goal of this work is to fill the spectral gaps between the bands and reconstruct a full spectrum from VIS to NIR, with a fine spectral resolution. Computing Eq. 2, with a dimensionality of $m = 26635$ is too computationally expensive. In order to reduce the dimensionality of this problem, as in Vidot and Borbás (2014), we apply a Principal Component Analysis (PCA) algorithm, which is an unsupervised

machine learning algorithm, and extract the principal components from the matrix $\mathbf{U}$.

    We need seven principal components (or eigenvectors) to solve our problem. As done by Vidot and Borbás (2014), we generate six principal components and we use a constant value for the seventh one, because this has been tested and shown to improve the performances. The other six principal components are generated from the three dry datasets described in the previous section. Since these datasets account for different surface types (vegetations spectra are only given in ECOSTRESS), and come

in different numbers, we cannot directly merge the spectra of the three datasets altogether. Thus, we balance the number of spectra from the different datasets clustering them using a k-means algorithm `sklearn.cluster.KMeans` (Pedregosa et al., 2011), as done in Liu et al. (2023). In this way, we obtain 100 representative soils spectra for the ICRAF–ISRIC and the LUCAS datasets each, and 128 representative spectra for the ECOSTRESS datasets extracted as follows: 40 vegetations spectra, 10 non-photosynthetic vegetations spectra, 40 soils spectra, 20 rocks spectra, 10 man-made materials spectra and 8

water bodies spectra. The water bodies spectra, which include snow of different granular sizes, frost, deep ocean, costal ocean and tap water, were not reduced in their dimensionality. Without accounting for this number difference, the vegetation and water surfaces present in the ECOSTRESS dataset would be outweighed by the number of soils spectra from the other datasets, resulting in a sensibly lower performance of the algorithm.

    We use the `scikitlearn.decomposition.PCA` implementation of PCA, which follows a Singular Value Decomposi-

tion (SVD) of the data as in Halko et al. (2009). From this process, we end up with a matrix $\tilde{U}_\lambda$ with the same spectral resolution as the laboratory spectra, where $\lambda$ represents the hyperspectral nature of this matrix. To combine it with the albedo data vector $\boldsymbol{R}$, which is only given at the seven MODIS bands, we need to convolve the full matrix $\tilde{\mathbf{U}}_\lambda$ with the average satellite response function of Terra and Aqua satellites of each band. This convolution is necessary to correctly estimate the measured albedo for the central wavelength of each band, crucial to generate the hyperspectral albedo maps with the PCA.

The result of the convolution is a square matrix $\tilde{\mathbf{U}}$ at the seven MODIS wavelengths available from satellites data. Since $\tilde{\mathbf{U}}$ is a square matrix, we can simply calculate

$$\boldsymbol{c} = \boldsymbol{R}\tilde{\mathbf{U}}^{-1}. \tag{3}$$

In this way, we have seven equations for seven coefficients, which allow us to estimate the coefficient vector $\boldsymbol{c}$. Once $\boldsymbol{c}$ is known, it is possible to calculate the albedo maps at all selected wavelengths as

$$\boldsymbol{R}_\lambda = \boldsymbol{c}\tilde{\mathbf{U}}_\lambda, \tag{4}$$

where the $\lambda$ subscript indicates the hyperspectral nature of the elements. The same process is applied to all the pixels in the map to generate a final albedo map with a spatial resolution of $0.05°$ in latitude and longitude, and to all the different days of the year, taking into account also Earth's seasonal variability.

    Vidot and Borbás (2014) created BRDFs maps using a PCA algorithm for their radiative transfer code. They use the ASTER

library (today called ECOSTRESS library), which at the time contained much fewer soils and vegetations spectra, to create average maps to include hyperspectral reflectivity of the soils in their radiative transfer simulations. Jiang and Fang (2019) demonstrated that increasing the sample of different soils from various countries of the world helps to validate several dataset

among each other. Without accounting for satellite data to create Earth's albedo maps, Jiang and Fang (2019) calculated eigenvectors, using a SVD algorithm, to study the hyperspectral properties of canopy trees in radiative transfer simulations, also

including small humid soils local datasets. For the scope of this work, it was not possible to directly use the three eigenvectors generated by Jiang and Fang (2019), as we regress the hyperspectral albedo maps from the seven MODIS bands, thus seven eigenvectors are needed.

As a result of the method explained above, we obtain an hyperspectral climatology of black-sky surface albedo over the

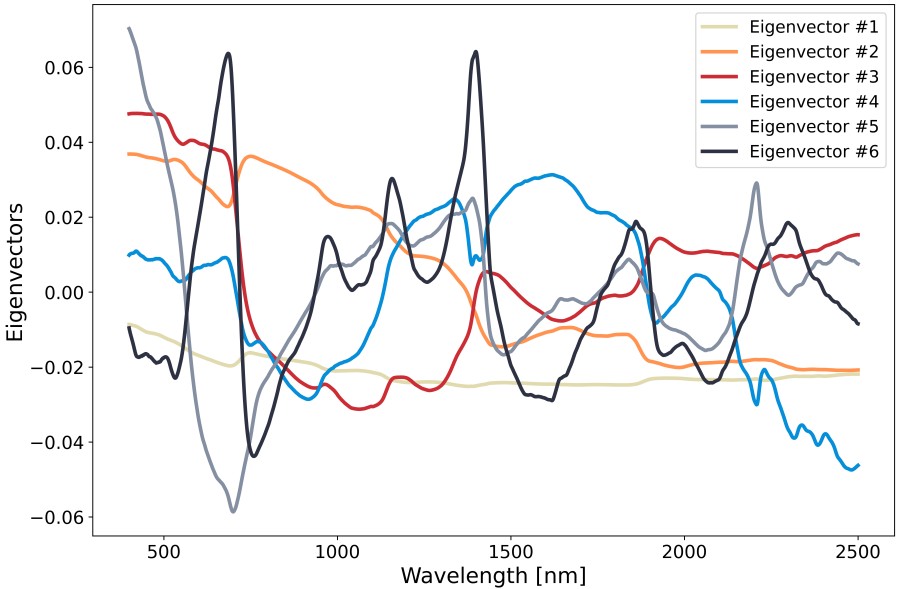

**Figure 3.** Eigenvectors generated by the PCA starting from the LUCAS, ICRAF–ISRIC and ECOSTRESS datasets. These eigenvectors are used to build the hyperspectral albedo maps. Eigenvectors are plotted in order of importance as calculated from the PCA.

entire globe from a wavelength range of 400 to 2500 nm in steps of 10 nm. While the interpolation is done with a 1 nm

resolution of the hyperspectral albedo maps, the final HAMSTER dataset has a spectral resolution of 10 nm to reduce the size of the single maps. We also reduce the spatial resolution of the hyperspectral albedo maps from the MCD43D 30 arc seconds resolution to 180 arc seconds, which correspond to 0.05° in latitude and longitude, again for size constraints. HAMSTER can be generated at the same spatial resolution of the MODIS MCD43D product and at a spectral resolution down to 1 nm, and higher spatial and spectral resolutions hyperspectral albedo maps are available upon request. The temporal resolution

of the hyperspectral climatology is of 1 day and it incorporates the information contained in the MODIS climatology and extends it to wavelengths that were not available before. HAMSTER is available at its finer spatial resolution (0.05° in latitude and longitude) at https://doi.org/10.57970/04zd8-7et52, while a coarser sparial resolution version, more suitable to global applications, is available at https://doi.org/10.5281/zenodo.11459410.

## 3 Validation

As a first test, we use the hyperspectral albedo maps to reconstruct the MODIS channels black-sky albedo of the climatology. We multiply the hyperspectral maps by the satellite spectral response function and we estimate the Root Mean Square Error (RMSE) for all the seven channels. For all MODIS channels (see Tab. 1), the RMSE is less than 0.0003. This confirms that the computed hyperspectral albedo maps are able to reconstruct the original MODIS climatology with great accuracy.

To validate the PCA retrieved maps (HAMSTER dataset), we compare them with the land surface albedo product of the SE-
255 VIRI instrument aboard the geostationary Meteosat Second Generation (MSG) satellite (Schmetz et al., 2002). SEVIRI has three channels in the VIS/NIR range, which are reported in Tab. 3. As MSG is a geostationary satellite, we cannot compare the entire world map, but only the Earth's "disk" that includes Africa and parts of Europe, South America and the Middle East. SEVIRI channels have spectral response functions that are broader than the analogous MODIS bands and are centred at slightly different wavelengths, and thus we convolved the hyperspectral maps to account for that. In particular, the SEVIRI channel
centred at 810 nm touches the vegetation "ramp" starting from 700 nm and is expected to show higher albedo values than the first SEVIRI channel.

The SEVIRI land surface albedo product MDAL (Geiger et al., 2008; Juncu et al., 2022, product identifier LSA-101) is of-

**Table 3.** Spectral bands of SEVIRI in the VIS and NIR providing information about the land surface. For each band, we specify the central wavelength and the bandwidth.

| band | central $\lambda$ [nm] | bandwidth [nm] |
|------|------------------------|----------------|
| 1    | 635                    | 600–680        |
| 2    | 810                    | 775–850        |
| 3    | 1640                   | 1550–1750      |

fered daily by the Land Surface Analysis (LSA) Satellite Application Facility (SAF) on the native SEVIRI grid with a spatial resolution of 3 km at the sub-satellite point, and is similar to the MODIS-based MCD43D product, against which it has been
evaluated (Carrer et al., 2010). As for MCD43D, both the bi-hemispherical (white-sky) and directional-hemispherical (black-sky) albedo are available. In order to compare with the HAMSTER hyperspectral albedo maps constructed from MODIS, we reprojected the SEVIRI data to the MCD43D grid, downscaled to 0.05° resolution in latitude and longitude, to allow for a consistent comparison. We selected two different DOYs, one in the late boreal winter (March 5th, DOY 065) and one in the middle boreal summer (July 30th, DOY 209) both in 2016 to compare the surface reflectivity during two different vegetation
stages with possible snow cover in winter and no snow in summer over Northern Europe. The results are shown in Figures 4 and 5.

We compare the three solar satellite channels offered by SEVIRI with the reconstructed channels from the HAMSTER climatology and HAMSTER single day (first three columns in Figs. 4 and 5). SEVIRI's channel 3 has the same central wavelength

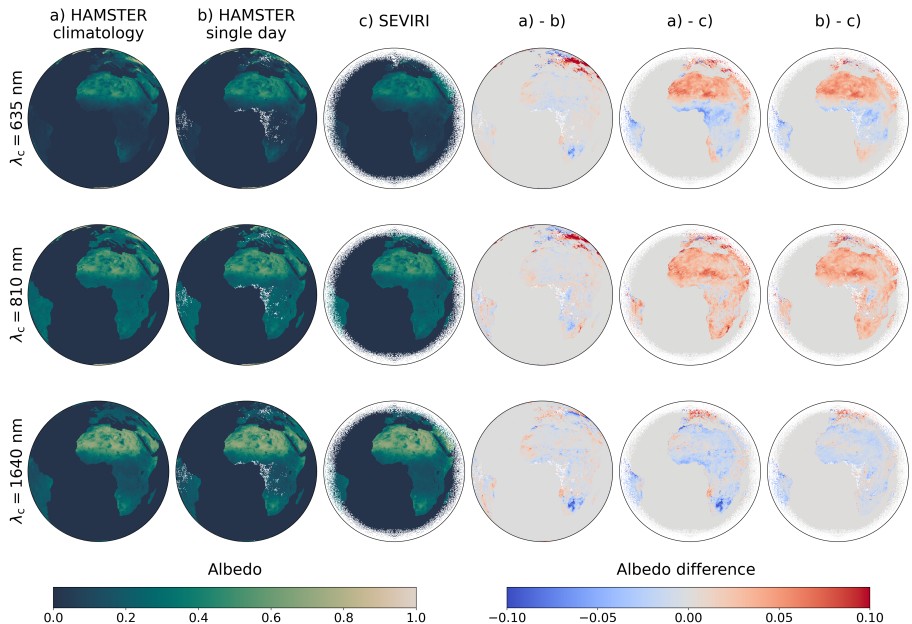

**Figure 4.** Comparison between HAMSTER climatology, HAMSTER single day and SEVIRI in the late boreal winter (March 5th 2016, DOY 065) for the three SEVIRI VIS/NIR channels. The first three columns show the albedo value for (a) the HAMSTER climatology and (b) the HAMSTER single day integrated over each SEVIRI channel, and (c) the SEVIRI albedo product. In the last three columns, we display the albedo difference between the three different albedo products or reconstructions between -0.10 to 0.10.

($\lambda_c$ = 1640 nm) as MODIS's band 6, which offers an almost direct comparison between MODIS and SEVIRI land surface
products. However, the hyperspectral nature of the retrieved HAMSTER maps is still used to convolve around the 1640 nm
MODIS band. The same happens for SEVIRI's channel 1 and MODIS's band 1, for which there is only a 10 nm difference
in the central wavelength. On the other hand, SEVIRI's channel 2 ($\lambda_c$ = 810 nm) is outside any MODIS band. This last case
allows us to make a comparison between the reconstructed albedo maps and the SEVIRI measurements, rather than between
the land surface products of the two instruments.
In addition, in Figures 4 and 5, we also assess the difference between the HAMSTER climatological average (first column)
and a single day HAMSTER reconstruction (second column), without accounting for the 10-year average of the climatology.
White pixels in the HAMSTER single day correspond to pixels without any albedo value from the MODIS MCD43D product.
The climatological average shows less features, in particular over Europe, which might be due to the fluctuations of a single
day, while in the HAMSTER single day we notice a larger dependence on the seasonality. The effect of the climatology is
shown in the forth column, where we plot the albedo difference between HAMSTER climatology and HAMSTER single day.
In Fig. 4 we clearly see discrepancies of the order of 0.10 in the first two channels, while in SEVIRI's channel three we notice
a lower albedo over Southern Africa for the HAMSTER climatology. Less differences are found for DOY 209, in the boreal
summer (Fig. 5). To conclude, the last two columns of Figs. 4 and 5 display the difference between HAMSTER (climatology

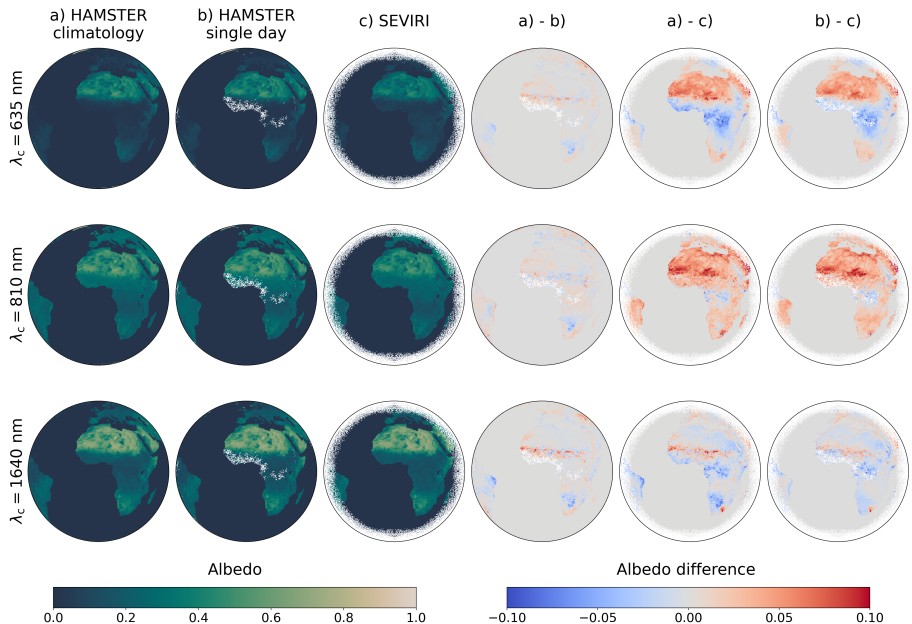

**Figure 5.** Comparison between HAMSTER climatology, HAMSTER single day and SEVIRI in the boreal summer (July 30th 2016, DOY 209) for the three SEVIRI VIS/NIR channels. The first three columns show the albedo value for (a) the HAMSTER climatology and (b) the HAMSTER single day integrated over each SEVIRI channel, and (c) the SEVIRI albedo product. In the last three columns, we display the albedo difference between the three different albedo products or reconstructions between -0.10 to 0.10.

and single day) integrated over the SEVIRI channels minus the SEVIRI land surface product. We notice an overestimation of the order of 0.05 for the reconstructed HAMSTER hysperspectral albedo maps in the first two channels over the Sahara desert, while vegetated areas over Africa and part of Europe and South America show a negative (SEVIRI channel 1) and positive (SEVIRI channel 2) discepancy compared to SEVIRI of around the same order. On the other hand, SEVIRI channel 3 ($\lambda_c$ = 1640 nm) is mostly underestimated by HAMSTER, with a smaller albedo difference compared to the other two channels. Since HAMSTER is built from the MODIS land surface product, our results are in accordance to the discrepancies found by Shao et al. (2021), which points towards difference between various land surface products of up to 0.06. While we describe the different offset arising from this comparison, we can conlude that the reconstructed maps are consistent, in their validation, to the discrepancies arising from different satellite data products.

In Figs. 6 and 7, we show the probability density function (pdf), calculated as the Kernel Density Estimation (KDE) (i.e., a Gaussian-kernel-based probability density, Scott, 1992) between HAMSTER (climatology and single day) and the SEVIRI land surface products for the two DOYs selected. For each comparison, we estimate the RMSE and we represent the discrepancies between the different albedo products with the KDE.

We notice that the RMSE is always very small, compatible with intrinsical differences between different retrieval of the albedo products. The RMSE is larger for SEVIRI channel 2 (centred at $\lambda_c$ = 810 nm) for both DOYs, which is also the SEVIRI chan-

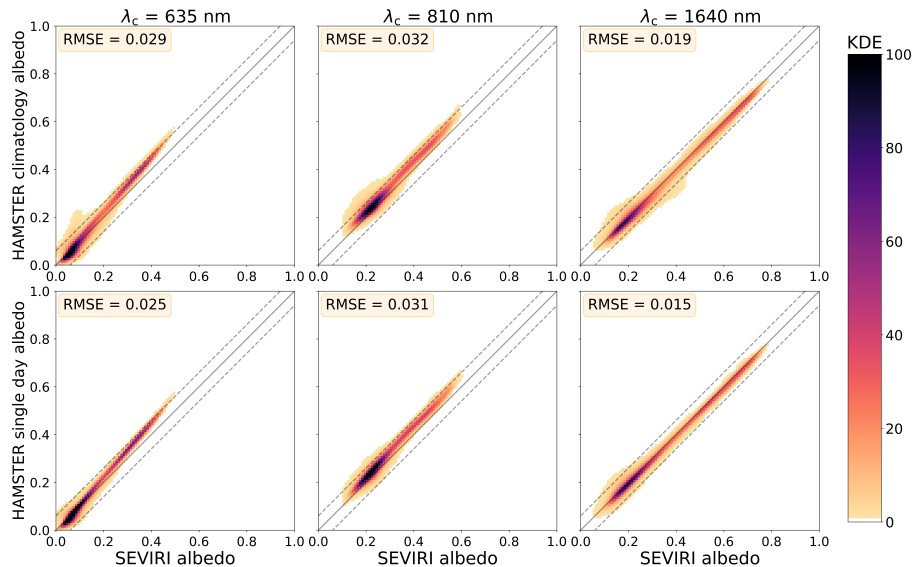

**Figure 6.** Kernel Density estimation (KDE) between HAMSTER climatology, HAMSTER single day and SEVIRI albedo data March 5th 2016 (DOY 065) for the three central wavelengths of SEVIRI channels (different columns). The first row accounts for HAMSTER climatology hyperspectral albedo maps, while the second row for the single day reconstruction. The soild line represent a perfect linear fit, while the dashed lines show linear fit with an offset of 0.06.

nel furthest from any MODIS channel. We also notice that the comparison with the hyperspectral maps built on the single day
albedos have always a slightly smaller RMSE, since the climatology can only reproduce a climatological vegetation state and
snow coverage pattern of a certain DOY.

In addition, we also calculate the RMSE between HAMSTER climatology and all three SEVIRI channels for each DOY is
2016 (Fig. 8). We can conclude that the two DOYs we selected for a more in depth analysis (DOY 065 and DOY 209) are rep-
resentative of the general trend. We notice that the comparison with SEVIRI channel 2 is, as expected, the one resulting in the
larger RMSE, being outside MODIS bands. However, the performace of the hyperspectral albedo maps are still in agreement
with the discrepances among different albedo products.

As a last test, we compare the hyperspectral albedo maps with the TROPOMI Lambertian Equivalent Reflectivity (LER)
product from https://www.temis.nl/surface/albedo/tropomi_ler.php (Tilstra et al., 2021, 2023). TROPOMI LER product (sub-
satellite pixel size of 0.125° x 0.125°) is remarkably different from MODIS MCD43D product, as it provides surface albedo
for snow/ice-free and snow/ice conditions separately. The snow/ice conditions are also averages over a month, which do not
allow for a direct comparison with MODIS, which provides daily snow coverages. Due to the high reflectivity of snow and
ice in the visible wavelengths, the large discrepancy among the two products does not come from the PCA retrieved albedo,
but from the different approaches in assessing the snow-coverage by the different products. On the other hand, TROPOMI
bands are very narrow, of just 1 nm, and they provide many channels in the Vegetation Red Edge (VRE) rump. For this reason,
we validate our hyperspectral albedo maps with the TROPOMI product only for the African continent and the Middle East,

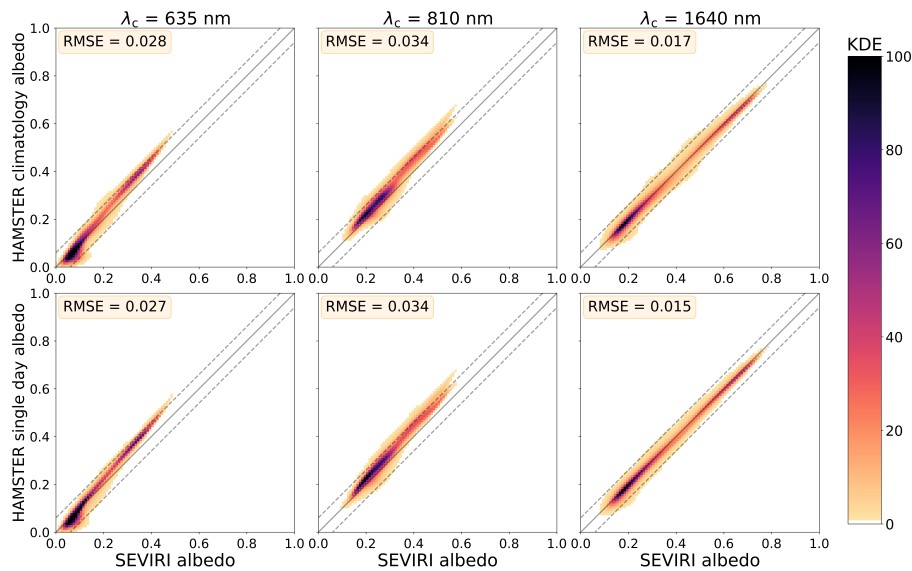

**Figure 7.** Kernel Density estimation (KDE) between HAMSTER climatology, HAMSTER single day and SEVIRI albedo data on July 30th 2016 (DOY 209) for the three central wavelengths of SEVIRI channels (different columns). The first row accounts for HAMSTER climatology hyperspectral albedo maps, while the second row for the single day reconstruction. The soild line represent a perfect linear fit, while the dashed lines show linear fit with an offset of 0.06.

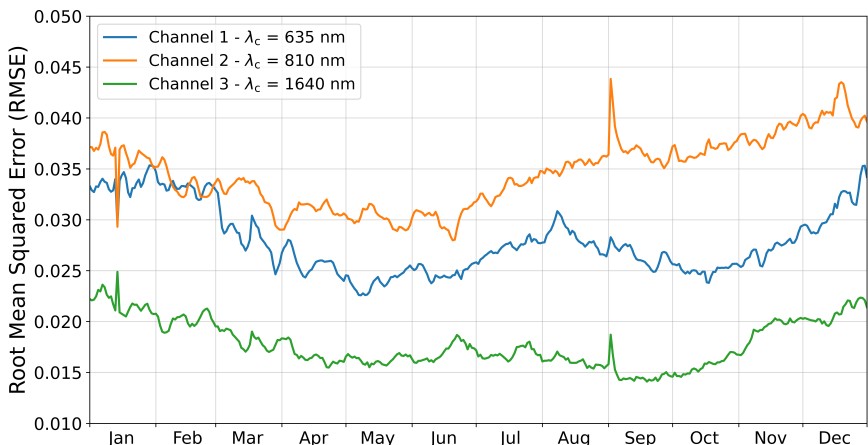

**Figure 8.** Root Mean Square Error (RMSE) of the comparison between HAMSTER climatology and all three SEVIRI channels. The comparison is done for each DOYs in 2016.

since this region exhibits the least snow coverage and allows for a direct and consistent comparison of land surface albedo among the two products. In this way, we avoid comparisons with snow/ice products which are not fully consistent. Due to the narrow satellite bands of TROPOMI, it was not needed to convolve for its satellite response function, and we estimated the

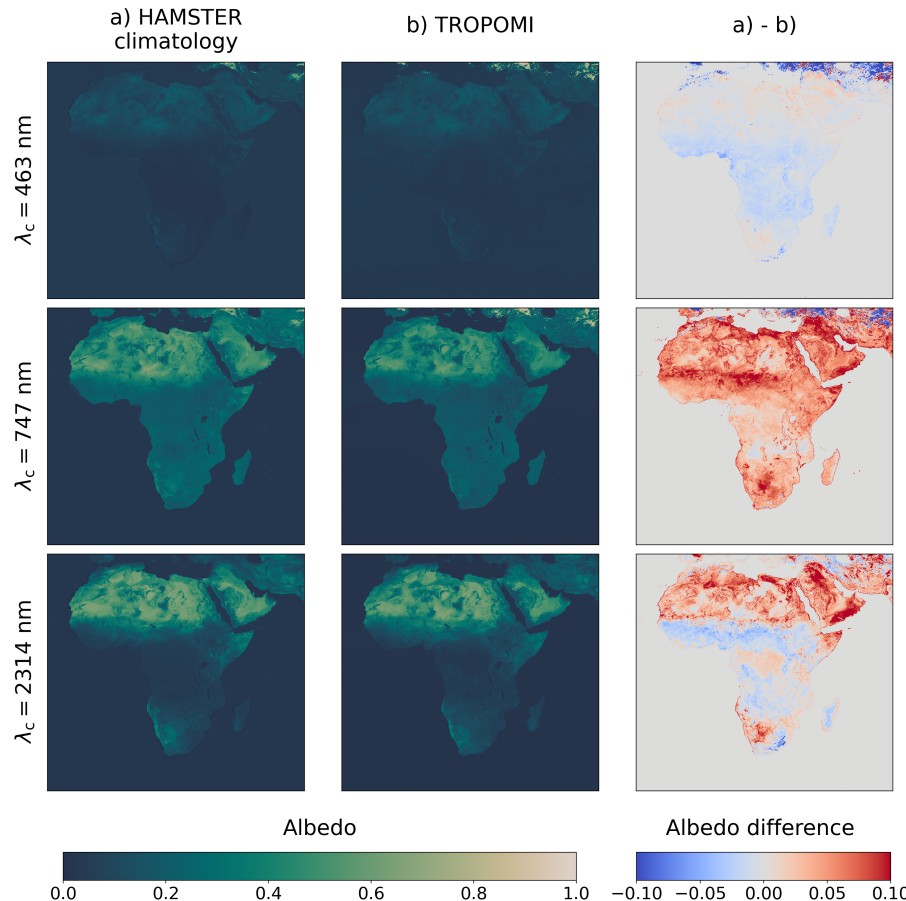

**Figure 9.** Comparison between HAMSTER climatology (first column) and TROPOMI in the late boreal winter (month of March) for three selected wavelengths among the TROPOMI VIS/NIR channels. The third column shows the albedo difference between HAMSTER climatology and TROPOMI LER albedo product.

RMSE between TROPOMI LER and our HAMSTER hyperspectral albedo product (at 1 nm spectral resolution). The results
are shown in Tab. 4. The RMSE is comparable to what we find for SEVIRI and with known discrepancies among different
surface albedo products, and it remains relatively small in the TROPOMI bands between 670 and 772 nm, inside the VRE
and far from MODIS bands. This confirms good performances of the hyperspectral albedo maps also far from the MODIS
bands on which they have been retrieved from. In Fig. 9, we select three TROPOMI bands and we compare the albedo value
over Africa between HAMSTER climatology (first column) and the TROPOMI albedo product (second column). We select the
TROPOMI monthly product for the month of March (average from 2018 to 2023), and we compare it with the average of the
HAMSTER from DOY 061 to DOY 091 (corresponding to all DOYs in March). In the third column, we again plot the albedo
difference between the two products. For $\lambda_c = 463$ nm, we notice a very good agreement, with discrepancies of around 0.019
over Africa. For $\lambda_c = 747$ nm, inside the VRE, the discrepancies are larger, with a general albedo overestimation of HAMSTER

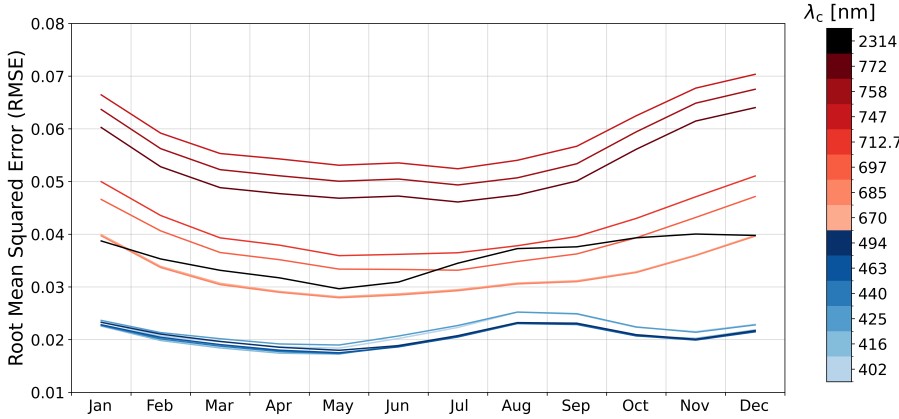

**Figure 10.** Root Mean Square Error (RMSE) of the comparison between HAMSTER climatology and all TROPOMI channels. The comparison is done for each month.

compared to TROPOMI, reaching differences of up to 0.10, but a overall RMSE of 0.055). We also compared the two products
with a band in the far NIR ($\lambda_c$ = 2314 nm) and we found an overestimation of dry and desert areas and an underestimation of vegetated regions in HAMSTER. Also in this last band, albedo products reach differences of up to 0.10, in particular over deserts, but with a small RMSE (0.033).

As done for SEVIRI, we also validate HAMSTER climatology against TROPOMI for each month, estimating the RMSE

**Table 4.** Spectral bands of TROPOMI LER product in the VIS and NIR, and RMSE of the comparison with the HAMSTER hyperspectral albedo maps over Africa.

| $\lambda$ [nm] | 402 | 416 | 425 | 440 | 463 | 494 | 670 | 685 | 697 | 712 | 747 | 758 | 772 | 2314 |
|---|---|---|---|---|---|---|---|---|---|---|---|---|---|---|
| RMSE | 0.019 | 0.018 | 0.020 | 0.019 | 0.019 | 0.020 | 0.031 | 0.030 | 0.037 | 0.039 | 0.055 | 0.052 | 0.049 | 0.033 |

for each TROPOMI band. Since TROPOMI offers monthly albedo products, we took the monthly averages of HAMSTER
climatology over Africa and the Middle East to perfom the comparison. In Fig. 10, we show the monthly validation results. For TROPIMI bands between 400 and 500 nm, the RMSE is always very small, of the order of 0.02. Moving into the VRE (from 700 to 800 nm), the RMSE is between 0.05 and 0.07, still comparable with discrepancies among different albedo products. For the NIR TROPOMI band ($\lambda_c$ = 2314 nm), the RMSE is of the order of 0.03-0.04 among all months.

## 4 Results

In this section we present the two main results of this paper: the MODIS black-sky surface albedo climatology for the seven bands and, building on that, the extended HAMSTER hyperspectral surface albedo dataset.

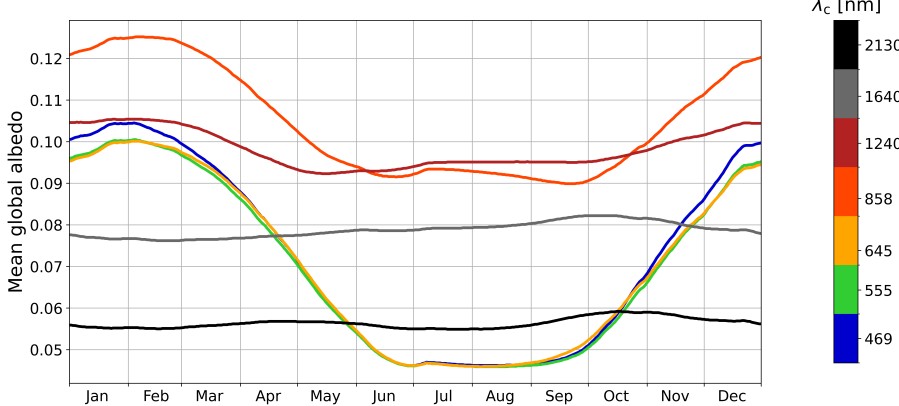

**Figure 11.** Yearly cycle of the MODIS climatology data black-sky albedo between 67°N and 67°S. The different curves represent the different MODIS channels indicated by their central wavelength.

## 4.1 MODIS climatology dataset

As described in Sect. 2.1, we have derived a 10-year climatology of surface albedo for different DOYs as a starting point to generate the hyperspectral albedo maps. This climatological average, with a temporal resolution of 1 day, allows studying the temporal variability of the albedo of the planet, as shown in Fig. 11. Since albedo values are not available for every pixel of Earth's surface during the year due to missing solar illumination during winter, we study the temporal evolution of the mean global albedo between 67° N and 67° S. Among these latitudes, we always have an estimate of the albedo of every single pixel for all DOYs. As a consequence, we are excluding from the mean albedo estimation both the Arctic and Antarctica regions, as well as other high latitude land surface in the Northern Hemisphere. For this reason, the mean albedo value should not be intended as a global estimate for Earth, but more as an indicator of its temporal variation.

In Fig. 11, we notice that the mean albedo is higher in the NIR bands, after the vegetation red edge (VRE) peaks. At 858 nm, which peaks right after the VRE, we notice the largest albedo value for the planet, followed by 1240 nm. Continuing in the NIR, with 1640 and 2130 nm, the albedo values decrease. On the contrary, in the VIS range there is a very small variation in the albedo among the three bands. The VIS bands show a clear seasonal trend due to the melting of ice and snow in the Northern Hemisphere, followed by a subsequent blossoming of vegetation. Thus, Earth's albedo is peaking in the late boreal winter in the VIS and then decreasing in the boreal summer. This large variability trend can be interpreted with seasonal differences in snow coverage, and it mainly follows the variability of the Northern Hemisphere since it host almost 80% of all the land in the globe. However, in the NIR bands, we notice other features around boreal late spring and autumn which are due to blossoming of flowers and reddening of leaves, which decrease the general reflectivity of green leaves.

In Fig. 12, we study the spatial variability of the albedo throughout the year for a particular wavelength for the entire 10 years climatological average. Here we selected MODIS band 1 centred at the 645 nm. In particular, we plot the difference between the maximum and minimum albedo values during the entire year, independently of when the maximum and minimum are

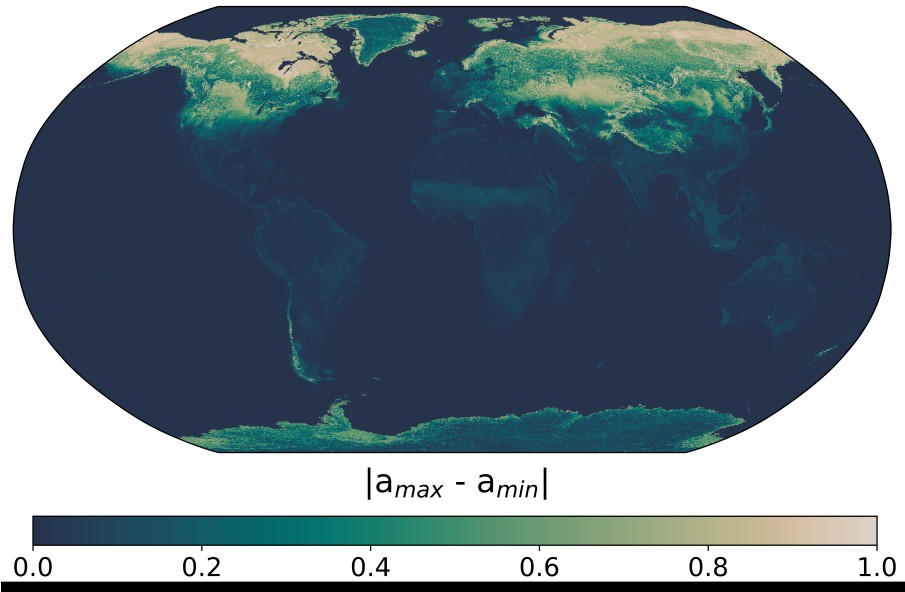

**Figure 12.** Spatial variation of the MODIS climatology, showing the difference between the maximum and minimum albedo value for each pixel during the year.

reached. For instance, the maximum of reflectivity over high latitudes in the Northern Hemisphere is reached during the boreal summer, while over the coast around Antarctica it happens during the austral summer, due to ice melting. It is important to note that the MCD43D product does not contain sea surface albedo product, and thus sea ice albedo. However, coastal regions exhibit albedo values and they are subject to large seasonal differences. Moreover, since albedo data are not available during boreal winter (summer) for the Northern (Southern) Hemisphere, for high latitude regions (north and south of 67°) the difference between the maximum and minimum albedo is calculated over a shorter time period, corresponding to the data coverage of the region.

Calculating this reflectivity variation for every pixel, the map in Fig. 12 highlights those regions that carry the largest variations. In particular, Arctic and Antarctic regions exhibit high reflectivity variations due to snow, ice and sea ice over coastal regions melting, clearly visible in the map. Mainland Greenland also shows more variability than mainland Antarctica, possibly pointing towards melting of Greenland's glaciers during boreal summer. Deserts all over the world, like the Sahara and Australian deserts, show the least variability, remaining almost constant throughout the year. Also, tropical rainforests, like the Amazon rainforest, do not exhibit a significant seasonal variability. On the contrary, temperate and boreal forests show a pronounced variation due to the difference between the snow cover during winter and the summer months.

## 4.2 Hyperspectral albedo maps

From the MODIS climatology data, we build the hyperspectral albedo maps with a PCA regression algorithm, as described in Sect. 2.3. The hyperspectral albedo maps allow us to combine the spectral features of different soils, vegetations and water surfaces with the high spatial and temporal resolution of the MODIS climatology data. This could potentially have many possible applications, from the implementation in climate models (where Braghiere et al. (2023) already demonstrated its feasibility) to the improvement of remote sensing retrieval frameworks. The new hyperspectral albedo maps have been implemented in the radiative transfer software package libRadtran (http://www.libradtran.org/doku.php, Mayer and Kylling (2005), Emde et al. (2016)).

As a first application, we use the hyperspectral maps to calculate the mean global albedo value close to the equinoxes. In this way, we have almost all pixels filled with an albedo value and it is possible to assess a mean albedo value for the entire globe as a function of wavelength (see Fig. 13). The main difference between the spring and autumn equinoxes relies on the snow coverage over the Northern Hemisphere, which increases the reflectivity during the spring boreal equinox. This affects mostly the VIS wavelengths, following the typical albedo profile of snow and frost (see Fig. 2). From these hyperspectral albedo maps, we could recover that the mean global albedo is around 0.21 in the VIS during March and around 0.17 over autumn, while it decreases below 0.10 in the NIR. The dots in Fig. 13 represent the average over the MODIS channels, without taking into account the hyperspectral albedo maps.

In addition, we apply the hyperspectral maps to study the VRE, which shows a steep increase in the reflectivity of vegetation due to chlorophyll, as shown in Fig. 13 at around 700 nm. In Fig. 14, we show the progression from 700 nm to 850 nm (with steps of 50 nm) of vegetation reflectivity for DOY 065 (March 5th). We notice a substantial increase of the albedo for all kinds of forests, from tropical to boreal ones, with the largest increase between 700 and 750 nm, as expected for the VRE. This comparison is only possible having albedo maps which account for their hyperspectral dimension. Using only the MODIS wavelengths, we would have missed the entire VRE transition because the closest bands are only at 645 and 858 nm.

As a last application, we study the spectral profile of different regions of the world, accounting also for their seasonal variability. We select different examples of rainforests, boreal forests, deserts, urban areas, and ice-covered regions as shown in Fig. 15. Inside the boundaries of the selected areas highlighted in Fig. 15, we average the spectra of all pixels in the region in order to obtain an average spectrum representative of the entire region. The averages are made for the four seasons separately. The first comparison pertains to forest spectra (dark green regions in Fig. 15). We selected three different rainforests, the Amazon, Borneo and Congo rainforests, two different boreal forests, over Canada and Russia, and a Savanna region over Kenya and Tanzania. The selection of the different areas is made by maximizing the possible land area hosting the same properties, but avoiding mixing the region with urbanized soils and different lands.

Fig. 16 shows the comparison between the spectrum of different forests. We notice a similar trend among all kind of forests, with similar spectral features. In particular, all forest shows three jumps in reflectivity of decreasing amplitude. The main difference between tropical rainforests and boreal forests resides, as expected, in their seasonal variability. Tropical rainforests do not exhibit almost any seasonal change, being very similar among each other. On the other hand, boreal forests experience an

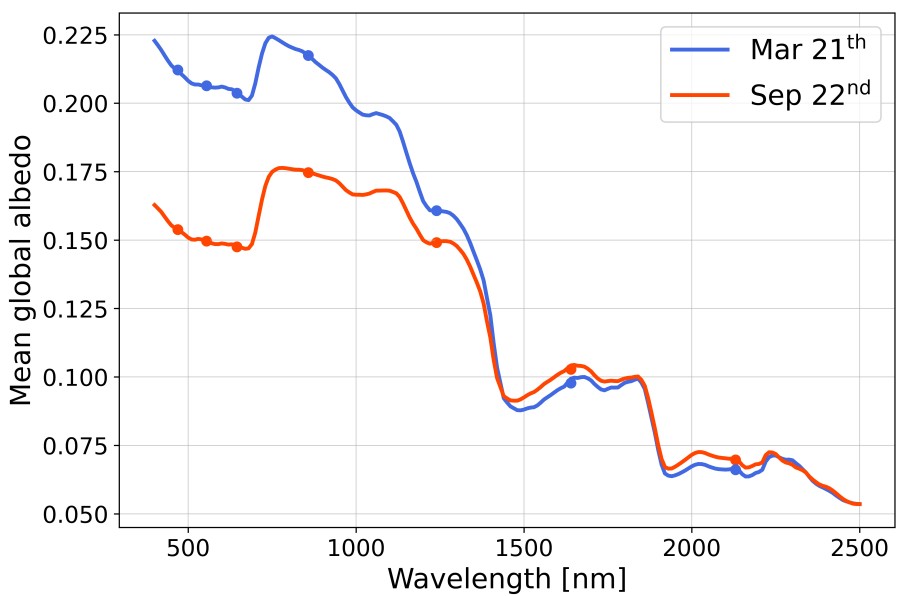

**Figure 13.** Mean global albedo as a function of wavelength over the entire globe. We select the two DOYs closest to the equinoxes, where we almost have all pixels filled with an albedo value. The seven dots show the albedo value of the seven MODIS bands, while the curves are built from the average of all pixels of the HAMSTER hyperspectral albedo maps for a given wavelength.

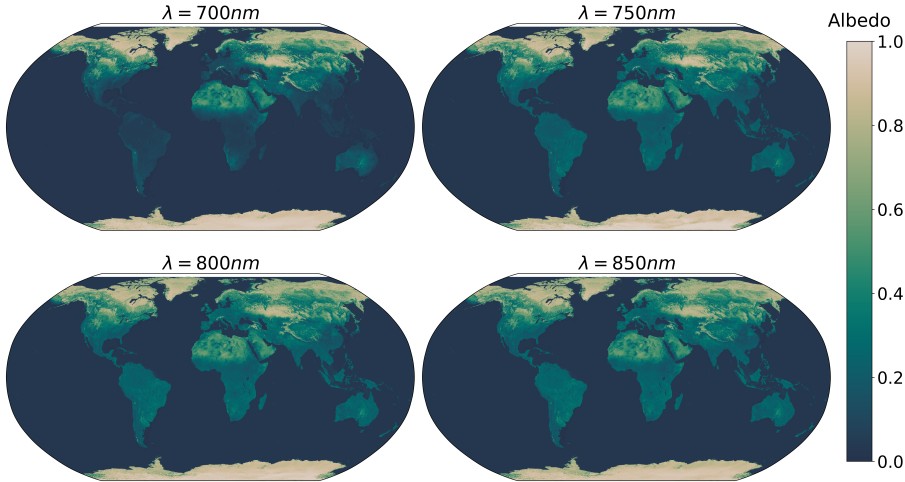

**Figure 14.** Spectral evolution of the surface albedo for March 5th (DOY 065). From $\lambda = 700$ nm to $\lambda = 850$ nm, there is a steep increase of the albedo over forests, due to the VRE.

important decrease of reflectivity from boreal winter to boreal summer. This is due to the melting of snow over boreal forests, which also happens with different timescales. There are also some small differences within tropical rainforests. Borneo shows

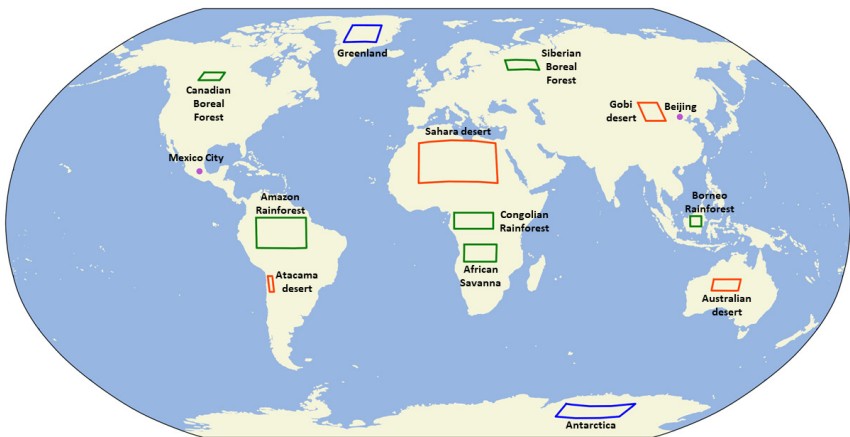

**Figure 15.** Regions of the world investigated. The green boxes represent the forests, the orange boxes the deserts, the blue boxes the ice sheets, and the purple circles the cities.

the least seasonal variation, while Congo shows the smallest reflectivity.

The final spectra are always combinations of different soils and vegetations, and the small differences we find are due to the different trees soils grounds and tree coverage of the different forests. If we compare the obtained spectra with Fig.2, we find an overall agreement with their main spectral features, but our final spectra are modulated by the combination of many different soils and averaged over seasons and different pixels.

We extend the comparison to desert areas (orange regions in Fig. 15). We select the Sahara desert, the Australian desert, the Gobi desert and the Atacama desert to extract the spectral properties from the hyperspectral albedo maps. Fig. 17 shows the comparison among different arid regions. We find that, among deserts, the reflectivity profile can greatly vary, depending on the mineralogy and composition of different soils and sands. In addition, as already discussed in Fig. 12, Sahara and Australian deserts do not display any significant seasonal change. This is not the case for the Gobi desert, which shows an enhanced reflectivity in the winter months, due to partial snow coverage.

In general, deserts exhibit common spectral shape, with a steep increase of reflectivity up to 750 nm, similar spectral features until the NIR, and a more or less steep decrease of reflectivity around 2150 nm. Different desert areas show larger discrepancies among themselves than forests.

The same methodology is applied to study the Greenland and Antarctica ice sheets (blue areas in Fig. 15). We select two regions which are always snow-covered to study their spectral features and seasonal patterns (see Fig. 18). As expected for fully snow covered surfaces, their reflectivity is very high, reaching almost 1 in the VIS, and it decreases in the NIR range. During Greenland and Antarctica's winters not all the pixels where always available, thus we averaged on less DOYs and less pixels to estimate their winter seasonal spectra. In Fig. 2, we see that snow and frost show different patterns in reflectivity, in particular in the NIR. This can explain the spread in the NIR spectra of both Antarctica and Greenland. This also needs to be combined

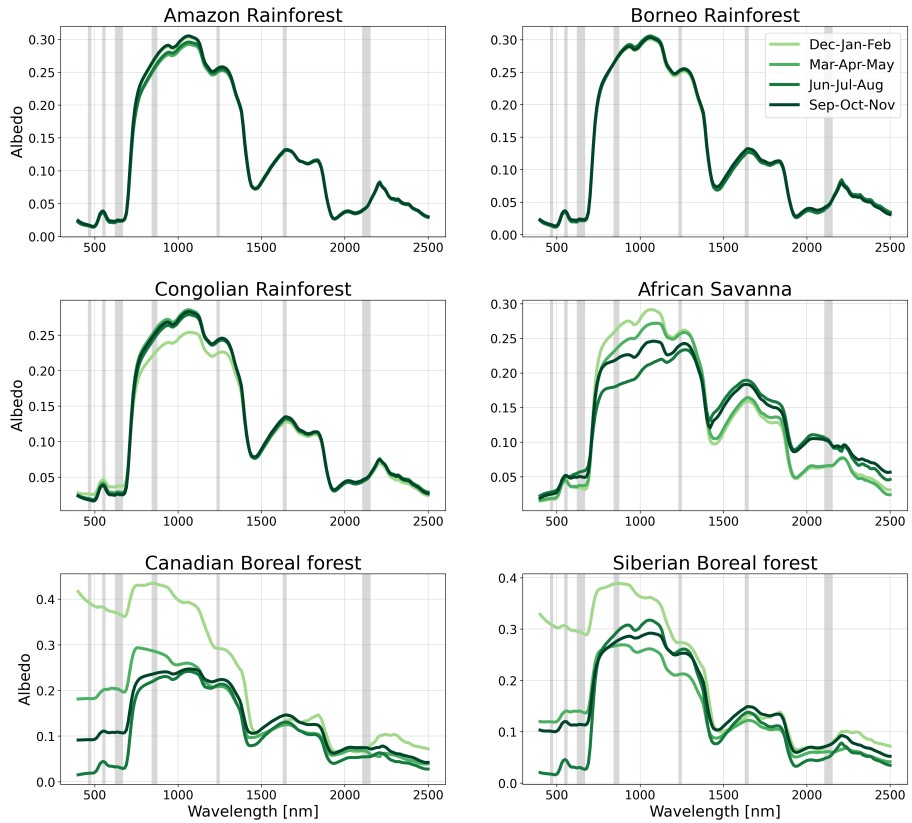

**Figure 16.** Spectra of different forests of the world obtained by averaging over all pixels in the corresponding region using the hyperspectral albedo maps. Seasonal variability is shown averaging the spectra over three-month periods, indicated with different colours. Gray bands represent the MODIS bandwidths.

with the formation of clear liquid water lakes on the surface of the glaciers during the melting season, which lowers the total reflectivity of the surface. For Greenland and Antarctica, we find similar behaviours in the NIR, with winter seasons exhibiting a higher reflectivity than summer seasons. We also notice that in the VIS there is almost no seasonal spectral variability over Antarctica, while Greenland shows two distinct trends between boreal autumn and winter and boreal spring and summer.

To conclude, we also extracted spectral profiles of two different cities: the urban areas of Beijing and Mexico City. Among the 45 man-made spectral material from ECOSTRESS, there are general construction materials, road materials, roofing materials and reflectance targets. Urban areas are treated as a linear combination of different components, like man-made materials, vegetation, and soils, and the PCA handles them as all other soils and vegetations spectra. MODIS albedo performances over cities has not been quantitatively assessed, and MODIS might underestimate surface reflectivity (Coddington et al., 2008), thus city spectra should be used with caution. Fig. 19 shows Beijing having a larger seasonal variability than Mexico City. In general, the spectra of the two cities look different, but with some common spectral features. Urban areas show a lower albedo than the other regions investigated, pointing towards the use of asphalt and concrete spectra in the PCA, and their general spectral

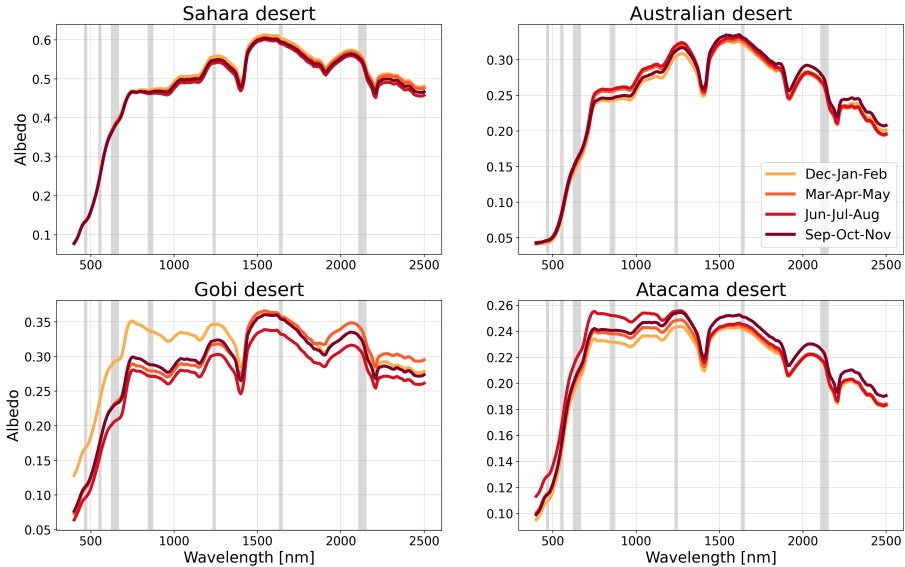

**Figure 17.** Spectra of different deserts of the world obtained from the average over different pixels from the hyperspectral albedo maps. Seasonal variability is shown averaging spectra over three-month periods, indicated with different colours. Gray bands represent the MODIS bandwiths.

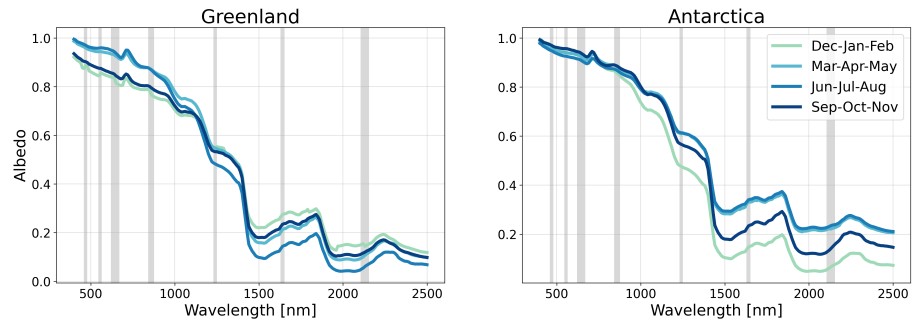

**Figure 18.** Spectra of different ice surfaces of the world obtained from the average over different pixels from the hyperspectral albedo maps. Seasonal variability is shown averaging the spectra over three-month periods, indicated with different colours. Gray bands represent the MODIS bandwiths.

shape appears different from all other regions. The steep increase in the VIS might be due to vegetation, while other features in the NIR come from man-made materials and different soils present in the training dataset. The peak of the reflectivity for urbanized areas is low, as expected. In general, extracting the spectra of different surface types, we found a good agreement among the typical spectral features of soils and vegetations expected to dominate the different surface types. For instance, different kind of forests all have the typical shape due to the VRE. However, the spectra of the various land types host much more information than the single spectrum of a tree or of a particular soil, and we can clearly see that they are a linear combination

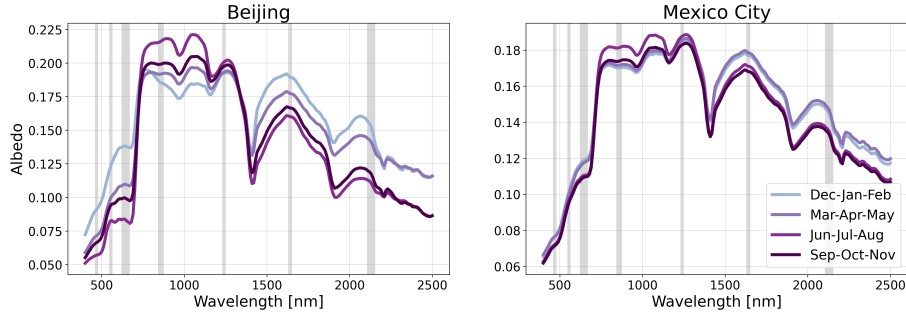

**Figure 19.** Spectra of two different cities (Beijing and Mexico City) of the world obtained from the average over different pixels from the hyperspectral albedo maps. Seasonal variability is shown averaging spectra over three-month periods, indicated with different colours. Gray bands represent the MODIS bandwidths.

of different spectra in the sample with varying weights. In fact, forests are the combination of trees with their typical spectral shape, but modulated by different soils reflectivities. As a result, the retrieved albedo of an entire forest is sensibly lower than the one of single trees in the dataset. This is in agreement with Jiang and Fang (2019), who generated different spectra for canopy tree radiative transfer simulations and studied the soils influence on the total reflectivity of the vegetated area. While typical vegetated features are always present in the spectrum, they are sensibly modulated depending on the properties of the background soil.

## 5 Conclusions

In this work, we create hyperspectral albedo maps to study the wavelength-dependent characteristics of the black-sky albedo of the Earth's surface. We select various soils, vegetations, snow, water bodies, and man-made materials spectra from three different datasets: the ECOSTRESS library, which has soils, vegetations, man-made materials, snow and water bodies spectra, LUCAS dataset, which contains different soils of many countries in the world, and the ICRAF–ISRIC dataset, a catalogue of thousands of soils of European Union countries. In total, we end up with 26635 spectra of different soils and vegetations from 82 countries of the world.

Due to the huge dimensionality of the final training dataset, we use a PCA regression algorithm to extract the principal components of the dataset. These principal components serve as eigenvectors to recover the albedo reflectivity of different pixels over Earth, starting from the MODIS land surface product. In particular, MODIS measures land surface properties in seven different bands in the VIS/NIR wavelength range. These seven MODIS bands are used as the starting point to build the hyperspectral albedo maps. With the PCA, we extract six principal components as in Vidot and Borbás (2014), and, with the addition of a seventh constant eigenvector, we combine them with the seven bands of MODIS data, for which the albedo of all single pixels is known. From this computation, it is possible to extract the spectral albedo value in the entire wavelength range pixel by pixel.

To generate climatological hyperspectral albedo maps, we use the 1 day land surface product from MODIS MCD43D product, and we average every DOY from 2013 to 2022. This allows us to get a climatological average of surface properties of the planet, to fill missing pixels which might be cloudy during a particular year, and to disentangle from the yearly variability patterns. As a final outcome, we obtain the HAMSTER hyperspectral albedo maps dataset with:

- a spectral resolution of 10 nm, in the range from 400 to 2500 nm;

- a spatial resolution of 0.05° in latitude and longitude;

- a temporal resolution of 1 day averaged in the time period between 2013 and 2022.

As demonstrated by Vidot and Borbás (2014) and Jiang and Fang (2019), PCA or SVD algorithms are powerful tools to combine a huge sample of soils and vegetations spectra and to reconstruct the albedo profile of different areas of the world. In our work, apart from generating hyperspectral albedo maps from the PCA as in Vidot and Borbás (2014), we also include Jiang and Fang (2019) advice to train the PCA with a much larger dataset, accounting for different countries of the world. In addition, our hyperspectral albedo maps are given for all 365 DOYs, thus making it possible to retain all the seasonal variability patterns present in MODIS's data.

Our MODIS climatological maps and hyperspectral albedo maps are validated against SEVIRI and TROPOMI land surface products. To perform this comparison, we adapted SEVIRI's dataset to MODIS projection, and we find that there is a good agreement between both MODIS climatology and the HAMSTER hyperspectral maps with SEVIRI observations, up to discrepancies of 0.06, which is a typical order of magnitude for land surface product comparisons (Zhang et al., 2010; Shao et al., 2021). Similar results are found in the comparison with TROPOMI.

Already the MODIS climatological dataset displays interesting temporal and spatial patterns. Thanks to both high spatial and temporal resolution, we study Earth's temporal variability for different wavelengths, and we display the maximal albedo difference of each pixel, highlighting regions with high temporal variability. The mean spectral albedo of the planet peaks at wavelegths larger than the VRE, while it shows a larger variability in the VIS wavelengths than the NIR ones, where the seasonal variation between snow covered high latitudes land in the Northern Hemisphere displays an increase of the surface albedo in the boreal winter.

We combine the information coming from temporal and spatial resolution of the MODIS climatology data with the possibility to spectrally extend the information about different regions to create typical spectra of different land surface type. We find that:

- forests, as expected, show typical vegetated spectral features, such as the VRE. Tropical rainforests do not undergo much seasonal change, while boreal forests have an increased reflectivity in the winter, when they get partially snow covered. Savanna regions experience a drying of the land after the end of the summer, which flatten the typical vegetation-induced spectral features.

- deserts show almost no seasonal variability, apart from those with occasional snow coverage. Depending on the properties of the soils, its colour and mineralogical composition, and the presence of sand, the overall reflectivity of the desert can greatly vary.

- ice and snow covered surfaces, like Greenland's and Antarctica's ice sheets, reflect almost entirely in the VIS, with a steep decrease in the NIR. During summer months, their albedo is slightly lower than late winter or spring months due to the melting of surface ice, which creates lakes on the top of the icy surface.

- urbanized areas, such as Beijing and Mexico City, are the combination of many different man-made materials, soils and vegetations spectra and their spectral shape host features from all of them. The total reflectivity of a city is lower than 20%.

These hyperspectral albedo maps dataset can be used for many different applications, from improving climate models to Earth's remote sensing, and to correctly simulate the disk-integrated spectra of Earth (Emde et al., 2017), and correctly model Earthshine observations (Sterzik et al., 2012, 2019). Only using the full spectral variations of land surfaces, it is possible to correctly establish Earth's energy budget. Braghiere et al. (2023) studied the impact of assuming only two broadband albedo values, as done in ESMs, versus using hyperspectral albedo maps. While the general radiative forcing is sensibly smaller than the one from doubling of $CO_2$, omitting the hyperspectral nature of Earth's surface causes deviation in many climatological patterns, such as precipitation and surface temperature, in particular over regional scales.

*Data availability.* The HAMSTER dataset is available at its finer spatial resolution (0.05° in latitude and longitude) at https://doi.org/10.57970/04zd8-7et52. A coarser spatial resolution (0.25° in latitude and longitude) and lighter version of HAMSTER, useful for global applications like ESMs simulations, is available on Zenodo at the following link: https://doi.org/10.5281/zenodo.11459410. The MODIS climatology used as the initial step to generate HAMSTER from the MODIS MCD43D product can be found at https://doi.org/10.57970/pt52a-nhm92. Finer spatial and spectral resolutions of the dataset, up to 30 arc seconds and 1 nm, respectively, are available upon request to the corresponding author.

*Video supplement.* A video supplement of this work is available at https://av.tib.eu/media/66248, where we show the spectral and spatial evolution of HAMSTER for four different DOYs.

*Author contributions.* GR designed the research. GR, LB and UH performed the MODIS climatology. GR and FG trained the PCA from the soils and vegetations spectra. GR performed the analysis and made the plots. GR, CE, MFS and MM interpreted the results. GR wrote the draft. All of the authors contributed in improving the manuscript. CW and CE implemented the dataset in libRadtran.

*Competing interests.* The authors declare no competing interests.

*Acknowledgements.* The authors thank NASA's MODIS/Terra+Aqua BRDF/Albedo Black-Sky Albedo Bands Daily L3 Global 30 ArcSec CMG MCD43D42-48 datasets for providing the albedo maps used in this work. We also acknowledge EUMETSAT for providing the SEVIRI dataset and ESA for the TROPOMI dataset. In addition, we thank the ECOSTRESS, LUCAS and ICRAF–ISRIC libraries for the surface spectra used in the PCA training.

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
