# Peer review of "HAMSTER: Hyperspectral Albedo Maps dataset with high Spatial and TEmporal Resolution"

_EGUsphere, 2024_

## Author Comment (AC1)

**Response to Comments by Reviewer #1**

We would like to thank the referee for the valuable comments on our manuscript. Please find below the answers to the issues. We report in blue the comments by the reviewer, and our answers can be found below each comment in black.

**Major issues**:

Choice of MODIS data: The MCD product user manuals quite clearly state that "*We recommend that users use the MCD43D 30 arcsecond products (or the 30 arcsecond MCD43GF gapfilled products) instead of the MCD43C products as much as possible, since the MCD43C products merely represent averages of the direct BRDF retrievals obtained with the MCD43D processing. The averaged MCD43C products thus necessarily include various QA flags within each pixel, and are therefore less rigorously high quality than the MCD43D products.*" Yet, the authors have chosen to base their analysis on MCD43C3. Why?

We thank the referee for this comment, and after carefully checking the different products, we decided to entirely redo the climatology starting from the MCD43D42-48 products. We changed accordingly all the description of the MODIS product in the manuscript, and its spatial and temporal resolutions. The entire analysis, the dataset, and all plots in the paper were done using the new product. We do not find any significant difference, but we notice better quality of mountain regions.

Since the MCD43D product is offered daily, we changed the temporal resolution of our dataset from 8 to 1 day. Section 2.1, where we describe our procedure to perform the climatology, was adapted to the new algorithm and analysis. As mentioned by the referee, this allowed us to obtain a better quality for our climatology.

We also included the daily climatology product on LMU Open Data (link in the Data Availability).

It is important to note that the spatial resolution offered by the MCD43D product is too large for our final HAMSTER dataset, which also includes the spectral component. All the steps of our analysis, from the climatology to the application of the PCA algorithm are done on the MCD43D product grid (30 arc seconds), and only the final HAMSTER dataset is downscaled on a 0.05° in latitude and longitude (180 arc seconds) for file dimension reasons.

Reference spectra: One is tempted to ask about the geographical distribution of the soil and vegetation spectra used here. For example, boreal forests are and behave quite differently from African rainforests – how well are the various regions of the Earth represented in the data? Do you have any means of assessing the uncertainty that is potentially related to misrepresented vegetation in particular, as the soil-vegetation references appeared to be heavily weighed towards soil sampling? What is the overall uncertainty of the HAMSTER data layers?

Vegetation spectra are only present in the ECOSTRESS library. It includes a collection of many different trees, shrubs and grass spectra, for a total of 487 spectra (line 140 in the manuscript) from the United States. Among trees, it includes both deciduous and evergreen trees and both broadleaves and needle leaved trees, from 'pinus lambertiana' to 'quercus douglasii' to 'eucalyptus ficifolia'. Among shrubs, we find from 'agave attenuata' to 'baccharis

pilularis' to 'salvia leucophylla' and many others. For grass, the dataset contains 'bromus diandrus', 'avena fatua' and normal 'grass'. Unfortunately, they do not specify the geographical distribution from which different vegetation spectra are collected. We would like to clarify that this is also the only dataset we found containing different trees and vegetation spectra.

This is different for soils spectra. The main reason to combine the three different spectra dataset (ECOSTRESS, LUCAS and ICRAF/ISRIC) comes from the need to collect spectra to cover a geographical distribution as broad as possible. While the ECOSTRESS library contains soils spectra only from the United States, the ICRAF/ISRIC dataset is a global dataset of soils spectra from 58 different countries and 785 sites (spanning Africa, Asia, Europe, North America, and South America), while LUCAS represents soils of the 28 European Union countries, including the United Kingdom. In total, our sample contains soil spectra from 82 different countries. Jiang and Fang (2019) noticed that the use of a global dataset significantly improves the accuracy of the hyperspectral soil reflectance modeling. Following their prescription, we put together one global (ICRAF/ISRIC), one continental (LUCAS) and one local (ECOSTRESS) dataset to reach the best possible accuracy in our PCA retrieval method. To our knowledge, ECOSTRESS is the only available dataset containing vegetation spectra, and thus it is not possible to assess how much vegetation might be misrepresented from a particular geographical region. We added two sentences about this in section 2.2, clearly stating this is a limitation of our approach.

We notice that our vegetation spectra are in agreement with the ones shown in Jiang an Fang (2019), where they simulated the spectra with a 3D canopy tree radiative transfer model which includes different soils as background.

The uncertainty of the HAMSTER data layers can only be assessed by its validation to other dataset, and this motivated us to validate it with SEVIRI and TROPOMI.

Treatment of snow within the HAMSTER maps: The reference spectra did not seem to contain snow and there was no mention of simulating snow spectra with snow models. How then did you achieve the PCA deconstruction over snow? What about mixed pixels with fractional snow cover and fractional vegetation cover, how are those handled?

The reference spectra contain snow measured spectra, as shown in Fig. 1, where we are plotting the 'coarse granular snow' spectrum from ECOSTRESS. In the ECOSTRESS library there are three different snow spectra: coarse granular snow, medium granular snow and fine snow. In addition, there are also frost and ice spectra, sea foam, sea water and tap water. All together, they form the 8 water and snow spectra from the ECOSTRESS library we mention in line 178 (new manuscript). The PCA handles snow as any other spectra, which means that for pixels with fractional snow and vegetation coverage the PCA can fit a linear combination of the most relevant trees, soils and snow spectra to better match the seven MODIS channels of the pixel. We included this detailed explanation in the manuscript.

Structure: Inclusion of HAMSTER validation in section on data and methods is confusing, as these are clearly results and not source data or its processing. Recommend revising the structure to start section 3 with the validation, and then proceeding towards analysis of features in HAMSTER data.

Thank you for the suggestion. We removed the Validation from the Data and Methods section. Now Data and Methods are section 2, Validation is section 3, and Results are

section 4. We kept the Validation and the Results in two separate sections, since the Validation is done comparing the climatology and HAMSTER with other albedo products, while in the Results section we analyze the different features found in the climatology and HAMSTER.

Gap filling on multiple layers: There are 5 different layers of gap filling needed – what are the respective percentages of global MODIS grid that are filled out at each of these steps? Do I understand correctly that in step 2, albedo values spanning to +/-40 (5x8) days of target date may be selected to fill the gap? 40 days during snowmelt or vegetation greenup could result in dramatically different gap-filled albedos relative to what was actually on the ground, what is the justification for this wide search window?

We included the statistics on the percentages of the gap filled pixels in the climatology (see new Fig.1). We now clearly describe how many pixels are filled at every step of the climatology. Even using the MCD43D product, to completely fill all pixels up to 90° of local solar noon zenith angle we need to move up to +-40 days (and beyond) from the actual DOY. Only a few pixels get filled with +- 40 days (only the very last horizontal layer above 80 degrees of local solar noon zenith angle), and this can be seen in Fig. 1. However, we still consider it very useful to fill all pixels among the local solar noon zenith angle for some possible applications, like Earth System Models and Earthshine simulations. We improved our climatological approach including a new step (step 3 in the new manuscript) to better represent snowmelt and green-up periods of the dataset. We never fill missing pixels with only values from the past or future DOYs, but we always average or linearly interpolate to avoid over or underestimation of snow coverage. We still believe this is a better assumption than using a default value for these high latitude regions, also being snow highly reflective in the visible. We included a better explanation of this in Section 2.1.
In the new climatology dataset uploaded on LMU Open Data, we also include a flag for every pixel which tells in which step the pixels have been filled. This way, users can also select to use the climatology with some missing pixels but without moving up to 40 days from the actual DOY.

**Minor issues**:

130 – I was under the impression that MCD products do not contain sea ice albedo or sea ice cover? Certainly the result figures show no Arctic sea ice, so why is sea ice referenced here and again in section 3? This also refers to the major issue on snow treatment – sea ice albedo is even more complex than snow, and I haven't seen anything in the manuscript about its treatment?

As the referee correctly points out, the MCD products do not contain sea ice albedo information because they do not provide sea surface albedo measurements. However, coastal regions are included in the MCD products and they show seasonal differences in surface albedo. We thank the referee for this comment, because we noticed that when we mention sea ice albedo in the manuscript this might sound misleading. For this reason, we rewrote the sentence in line 130 (old manuscript) to expand our explanation. In the results, we particularly discuss Greenland and we notice a clear difference between boreal summer and boreal winter, where sea ice albedo shows a huge difference for pixels surrounding

Greenland, and these pixels are among the ones exhibiting the largest variability in the dataset.

Fig 7 – TROPOMI coverage extends elsewhere from Africa, why only focus the comparison there?

As mentioned in line 273, the TROPOMI LER product is very different from the MODIS product since it provides surface albedo for snow/ice-free and snow/ice conditions separately. In particular, including the snow/ice conditions provided by TROPOMI, we found large differences in snow coverage, with the TROPOMI product showing substantially more extended snow coverage over the Northern Hemisphere (DOY 065, March 5th 2016). This is because the snow/ice conditions in the TROPOMI LER products do not represent the daily coverage (as in MODIS) but averages over a month. Due to the high reflectivity of snow and ice, this does not allow for a direct comparison of the two products, in particular in the visible wavelengths. The large discrepancy among the two products does not come from the PCA retrieved albedo in HAMSTER, but from the different approaches in assessing the snow-coverage by the different products. For this reason, we selected a region over Africa and the Middle East to validate HAMSTER with TROPOMI, since this region exhibits the least snow coverage and allows for a direct and consistent comparison of land surface albedo among the two products.
In the manuscript, we included a more detailed explanation to clarify why we perform the validation with TROPOMI only over Africa and the Middle East.

Section 3 – the PCA-interpolated spectral albedos appear reasonable for most land surfaces, certainly. But I question the validity of the urban areas' spectra, there did not seem to be any reference spectra on man-made structures in the study? And how would PCA handle the extremely nonlinear shifts in land cover and surface material that are common to urban areas? Since the overall quality of MODIS albedos over cities has not been quantitatively assessed and the validity of the RossThick-LiSparse retrieval is uncertain over them, I would be very careful of highlighting those areas in particular unless the authors can prove that their spectra are valid.

In the ECOSTRESS library there are 45 man-made material spectra, as mentioned in line 140 (old manuscript) and 165 (new manuscript). Among these 45 spectra, there are general construction materials (construction concrete, black gloss paint, pine wood, red smooth-faced brick, etc.), road materials (construction asphalt, construction tar, etc.), roofing materials (like copper metal and terra cotta tiles) and reflectance targets (brass plate). The PCA is able to treat these spectra as all other spectra present in the datasets. Urban areas are a linear combination of different components, like man-made materials, vegetation, and soils. This is similar to other regions of the world, such as forests or deserts, which are a linear combination of many different soils, rocks, minerals and vegetation, and are handled by the PCA in the exact same way.
Overall, the spectra of urban areas appear reasonable. They show a lower albedo than the other regions investigated, pointing towards the use of asphalt and concrete spectra in the PCA, and they show some of the features coming from forests, but the general spectral shape appears different from all other regions. We described the following in lines 444-448. We also include a cautionary remark mentioning that MODIS albedo performances for urban areas have not been assessed yet.

---

## Author Comment (AC2)

**Response to Comments by Reviewer #2**

We thank the referee and appreciate the helpful comments to improve our paper. We have addressed them in the latest version of the draft and reply below. The modifications are in bold font face in the updated manuscript version.

Please note that, as suggested by the other referee, the entire analysis was done again changing the initial MODIS albedo product from MCD43C3 to MCD43D42-48. The current MODIS product we are using has a better spatial resolution and is of higher quality. All references to the MODIS product, its temporal and spatial resolutions in the paper were changed accordingly. Overall, the results did not change, and the conclusions of the paper remain the same.

**Major Comments:**

M1. For the seven climatology calculation steps, you need to specify/quantify how many missing pixels remain before/after each step, preferably as a percentage.

As suggested by the other anonymous reviewer, we have redone the climatology with the MCD43D42 MODIS product, which has a better spatial resolution and allows for an overall better quality climatology. As a consequence, we adapted section 2.1 to the new analysis, and we included the percentage of filled missing pixels of the climatology at each step in the item list. We also included a new figure (Fig. 1) to show the differences of missing pixels at each step of the climatology averaging over all DOYs.

M2. In light of Figure 3 and 4, it becomes evident that the employed method did not successfully generate accurate albedo representations for spectral bands not covered by the input dataset. The primary objective of this study is to construct an almost continuous albedo dataset using a limited number of input bands. While the direct comparison for the 1640 nm band exhibits strong agreement, the 810 nm band—outside the MODIS bands—shows the poorest comparison. However, it is precisely in the bands excluded from MODIS that the value of the HAMSTER product becomes apparent. Am I missing something here?

HAMSTER was trained on the 7 available MODIS bands in the VIS and NIR which carry information on the surface, thus it reproduces them exactly. It is expected that HAMSTER performs better in reconstructing albedo values for other albedo products in the wavelength bands closer to MODIS bands. The 1640 nm SEVIRI comparison shows the best performance of HAMSTER compared to SEVIRI, considering the RMSE as the metric to assess the performance. This was also expected because MODIS band 6 is centered at the same wavelength as SEVIRI channel 3. However, the referee mentions only the comparison with SEVIRI between 1640 nm and 810 nm (which is a SEVIRI band outside MODIS bands). If we take into account the other SEVIRI band we are comparing (635 nm), this is also almost centered at a MODIS band (645 nm, with bandwidth 600-680 nm, table 3), and HAMSTER basically performs the same in the validation with SEVIRI between 635 nm and 810 nm, being 810 nm very far from any MODIS band. From this, we can conclude that HAMSTER is doing well in representing albedo products outside the bands it was trained from, and the differences found are mostly intrinsic to the different albedo products from different satellites. In the paper we mention many studies assessing large discrepancies among different albedo products. We should also consider that the total reflectivity of the

planet is lower in the NIR (1640 nm), so we expect less variability here and an easier comparison among different albedo products.

In addition, the referee should not focus on Figure 3 and 4 to assess the performance of HAMSTER, because Figure 3 and 4 are meant for spatially assessing where albedo is over/underestimated in the comparison. The referee should refer to Figure 5 and 6 to look at the RMSE scores. Between 635 nm (MODIS band) and 810 nm (outside MODIS band), the RMSE goes from 0.029 to 0.032, which means that HAMSTER is successfully reconstructing albedo values outside the bands it was trained. It is true that for 1640 nm the RMSE is even better (0.019), but as we mentioned before, in the NIR there is less spread because the surface is less reflective in general. 810 nm is in the middle of the vegetation red edge ramp, so it is the most reflective region in the entire VIS/NIR range, and it makes sense the spread is larger.

M3. In Figures 5 and 6, the comparison is limited to only two specific days. While the separation into winter and summer is understandable, it raises the question of why a more extensive set of days was not utilised for potentially robust comparisons. Given the availability of numerous days, it would be beneficial to assess whether these two selected days are truly representative of broader trends.

We thank the referee for this comment, which helped us to fully address the performance of the validation of HAMSTER compared to SEVIRI and TROPOMI. We included two new figures in the manuscript (Figs. 8 and 10) where we compared each day of 2016 between HAMSTER climatology and SEVIRI (Fig. 8) and the monthly difference between HAMSTER climatology and TROPOMI (Fig. 10). TROPOMI only offers monthly products, thus we did monthly averages of HAMSTER to perform this comparison. The comparisons are done for all SEVIRI and TROPOMI channels and bands and report the RMSE. We note that there is no large trend and the two selected days (DOY 065 and DOY 209), which are studied in greater detail, are representative of the general trend.

**Minor Comments:**

L32: The word "the" is missing between "compared" and "Moderate"

Solved

L34: Which specific land surface products are being referred to?

The land surfaces products mentioned in Shao et al. (2021) are MODIS, GLASS (Global LAnd Surface Satellite), and CGLS (Copernicus Global Land Service). We included them in the sentence.

L42: Please define the acronym VIS upon first use.

Solved

L52: I would argue that radiative transfer simulations do not demand all wavelengths, but only the representative ones. The simulations likely does benefit from higher resolution. Could you support your statement with a reference?

We included as a reference Li and Yang (2024), which shows how retrieval of the cloud pressure thickness from the O2A band is dependent on the albedo assumption overlaying the region. Thus, to correctly perform retrievals, you need precise radiative transfer calculations across all wavelengths.

L60: Not sure what you mean by extrapolated in time, could you describe it a bit more.

We changed the sentence. We just mean that we build a climatology (10-year average with subsequent filling of missing pixels as described in section 2.1) for each DOY, since the MCD43D product provides albedo data for each DOY separately.

L70: Consider adding a sentence to differentiate your work on hyperspectral albedo maps from that of Braghiere et al., emphasising any novel aspects.

We included a short paragraph explaining Braghiere et al. (2023) approach to build an hyperspectral soil reflectance map to our dataset.

L96-100: Could you provide a clearer description or explanation for the content in lines 96-100? It was challenging to grasp.

We rewrote the paragraph, also taking into account the difference from the previous climatology (8 days of temporal resolution) with the climatology obtained from the MCD43D product (1 day of temporal resolution). Also the spatial resolution among the two products is very different, and this was adjusted in the manuscript.

L124: Could you explain the rationale behind assuming no atmosphere in this part?

We need to assume no atmosphere, otherwise we would not treat the reflection of the ocean surface anymore, but rather the reflectance of the ocean plus Earth's atmosphere, including atmospheric absorption band features.

L335-336: Please verify the parentheses in this section to ensure correctness.

Solved

L347: Could you provide additional context or highlight the substantial increase mentioned?

We had a sentence referring to the increase at around 700 nm as seen in Fig. 10.

L408: Please specify the number of principal components obtained in your analysis.

It is already written two lines below "With the PCA, we extract six principal components as in Vidot & Borbas (2014), and, with the addition of a seventh constant eigenvector, we combine them with the seven bands of MODIS data" (L411-412, old manuscript)

L419: Include the range here.

We included the range.

L432: Consider adding a statement explaining that performance deteriorates as wavelengths move farther away from the MODIS input bands (as mentioned in comment M2).

As explained for comment M2, performances do not substantially deteriorate going outside MODIS bands. Since outside the MODIS bands we are interpolating with the PCA, it is expected that performances will slightly deteriorate, and we do not think it is relevant to mention it again at this stage.